# Dynamic Schwartz-Fourier Neural Operator for Enhanced Expressive Power

**Wenhan Gao**                                                    *wenhan.gao@stonybrook.edu*
*Stony Brook University*

**Jian Luo**                                                      *jian.luo@stonybrook.edu*
*Stony Brook University*

**Ruichen Xu**                                                    *ruichen.xu@stonybrook.edu*
*Stony Brook University*

**Yi Liu**                                                        *yi.liu.4@stonybrook.edu*
*Stony Brook University*

**Reviewed on OpenReview:** *https://openreview.net/forum?id=BOE2yjrNb8*

## Abstract

Recently, neural operators have emerged as a prevailing approach for learning discretization-invariant mappings between function spaces. A particular example is the Fourier Neural Operator (FNO), which constrains integral kernels to be convolutions and learns the kernel directly in the frequency domain. Due to the capacity of Fourier transforms to effectively reduce the dimensionality and preserve information, FNOs demonstrate superior performance in terms of both efficiency and accuracy. In FNOs, the convolution kernel is fixed as a point-wise multiplication in the frequency domain; however, these translation-invariant kernels might limit the expressiveness of FNOs. For instance, if the underlying system lacks translational symmetries, the kernels learned by the FNO will still exhibit translational invariance, thereby limiting the model's expressive power. We propose a dynamic Schwartz operator that induces interactions between modes to enhance the expressiveness of FNOs. In this work, we introduce a novel approach that equips FNOs with Schwartz operators to learn dynamic kernels, termed Dynamic Kernel Fourier Neural Operators (DSFNOs). By incorporating this dynamic mechanism, our model gains the ability to capture relevant frequency information patterns, facilitating a better understanding and representation of complex physical phenomena. Through experiments, we demonstrate that DSFNOs can improve FNOs on a range of tasks, highlighting the effectiveness of our proposed approach. The code is available at `https://github.com/wenhangao21/TMLR25_DSFNO`.

## 1 Introduction

In various areas of science and engineering, researchers seek to explore how physical systems behave under varying conditions, such as different initial conditions, boundary values, or forcing functions. Traditional numerical methods, such as finite difference, finite volume, finite element, and spectral methods, frequently require excessive time for simulating physical systems with varying parameters. A type of data-driven surrogate model, termed neural operators (NOs), serves as an efficient alternative (Kovachki et al., 2023; Zhang et al., 2025). An operator learner aims to learn a mapping between the parameter function space and the solution function space. A well-trained neural operator can evaluate different parameter functions, such as different initial conditions, without requiring re-training. Evaluation to a parameter function only necessitates a forward pass of the network, which can be several orders of magnitude faster than traditional numerical methods (Li et al., 2021). Pioneering works of neural operators include, but are not limited to, DeepONet (Lu et al., 2021), which is built upon the universal approximation theorem of operators;

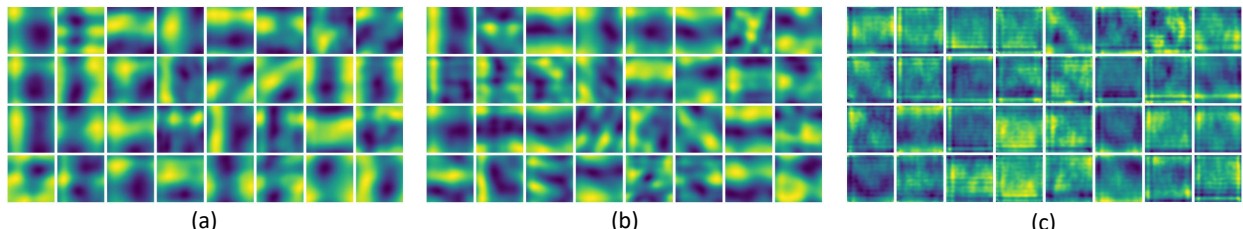

(a)                                    (b)                                    (c)

Figure 1: Learned filters for FNO and DSFNO on the Darcy flow equation: (a) Learned DSFNO filters for input function $a_1$; (b) Learned DSFNO filters for input function $a_2$; (c) Learned FNO filters which are fixed for all functions. The 32 input channel filters in the first kernel integral layer are presented. It can be observed that DSFNO filters can focus on different local details based on different input functions, with bright and dark colors being more distinct and clearly separable. In contrast, since FNO filters are fixed for the entire dataset, they tend to be more global for each filter and do not capture the distinct local details for different input functions as effectively. Additional visualization is provided in Appendix A.

GNO (Anandkumar et al., 2019), which parametrizes integral operators with message passing; and Model Reduction (Bhattacharya et al., 2021), which employs model reduction (PCA) to learn mappings between infinite-dimensional spaces. Another prominent and influential example of neural operators is the Fourier Neural Operator (FNO) (Li et al., 2021) and its variants (Rahman et al., 2023; Tran et al., 2023; Bonev et al., 2023), which model the solution operator as a composition of kernel integral operators and non-linearities and impose the integral kernel operator as a convolution operator. FNO uses the Convolution Theorem to learn the kernel directly in the frequency domain. Owing to the capacity of spectral analysis to capture information in a discretization-invariant manner while reducing the dimensionality of the problem, FNO has demonstrated its capacity to serve as an efficient and accurate surrogate model for physical modeling and has been applied to various fields (Pathak et al., 2022; Chen et al., 2023; Jiang et al., 2021).

Kernel integral operators, including FNO, are motivated by the Green's function or fundamental solution of PDEs (Kovachki et al., 2023). The Green's function kernel used to express solutions of PDEs can depend on the input parameter function (Evans, 2010); under fairly general conditions, we can express the solution, $u(x)$ of a linear PDE as:

$$u(x) = \int_\Omega G_a(x, y)f(y)dy, \tag{1}$$

where the integral kernel $G_a$ can be dependent on the parametric function $a(x)$. In other words, **if the parameter function changes, the integral kernel changes as well. In contrast, in the formulation of integral neural operators, once trained, the kernel is usually fixed for all input functions.** It motivates us to study dynamic kernels that adapt to the input parameter functions for integral neural operators. Moreover, FNO additionally constrains the kernel to be a convolution kernel to effectively harness the power of Fast Fourier transforms to model the kernel integral; however, this constraint further limits the learning capacity of the model. A simple example is when the underlying system lacks translational symmetries; in such cases, the kernels learned by FNO will still exhibit translational invariance, limiting its expressive power. To address these challenges, we propose a Dynamic Schwartz-Fourier Neural Operator (DSFNO) that utilizes dynamic Schwartz operators to learn convolution kernels to augment the expressive power of FNOs. Examples of learned dynamic kernels (Fourier filters) are provided in Figure 1; (a) and (b) are the learned kernels for different input functions that change as the input changes; (c) is the learned fixed FNO kernel. **In summary, our contributions can be summarized as follows:** 1) We highlight a critical yet often overlooked issue concerning the limited expressive power of fixed convolution kernels; 2) We propose a novel general framework to learn dynamic kernels to capture more diverse and spatially-dependent dynamics; 3) We propose Schwartz operators that represent general linear operators between function spaces while facilitating dimensionality reduction for efficiency.

## 2 Related Work

In recent years, neural PDE solvers have gained much attention as a promising alternative to traditional numerical methods for tackling PDE challenges in a wide array of practical engineering and life science

applications, including climate modeling (Pathak et al., 2022), fluid dynamics (Zhang et al., 2022; Sirignano et al., 2020), material science (Kumar & Echekki, 2024), *etc.*. Traditionally, the process of solving a PDE entails finding a smooth function that satisfies the derivative constraints presented by the equations; many traditional numerical solvers are developed based on this view, such as the Finite Difference Method (Quarteroni & Valli, 1994). From this viewpoint, Physics-Informed Neural Networks (PINNs) (Raissi et al., 2019; Li et al., 2022; Gao & Wang, 2023; Wang et al., 2021) have emerged to approximate solutions of PDEs. However, PINNs approximate PDE solutions on an instance-by-instance basis; each time the parameter function changes, PINNs need to be retrained. Another perspective from modern functional analysis is to view differential operators as mappings between function spaces, then the inverses of these mappings naturally serve as the solution operators. Expanding on this perspective, neural operators have been introduced as a technique for addressing families of parametric PDEs (Li et al., 2021; Lu et al., 2021; Kovachki et al., 2023; Brandstetter et al., 2022; Raonic et al., 2023; Bhattacharya et al., 2021; Gao et al., 2024). After training, inference on a new instance of the parameter function only requires a forward pass of the network, which can be notably faster, often by several orders of magnitude, compared to traditional numerical methods.

Among neural operators, Fourier neural operators have garnered considerable attention and popularity. FNO is a type of iterative kernel integral solver, termed as integral neural operators, and the kernel integral is imposed to be a convolution (Kovachki et al., 2023). Global convolution takes place through Fourier layers in the frequency domain; by utilizing low-frequency modes and truncating high-frequency modes, FNO captures global information while incurring low computational costs. Since FNO serves as the backbone in many models designed for specific applications (Pathak et al., 2022; Chen et al., 2023; Jiang et al., 2021; Bonev et al., 2023), it is imperative to enhance FNO itself. Lately, several studies have focused on enhancing the learning capabilities of FNO. Zhao et al. (2023); Liu-Schiaffini et al. (2024) enable FNO to incorporate local receptive fields, allowing for the retention of high-frequency local details. Liu et al. (2023a) expands the application of FNO to non-rectangular domains. Liu et al. (2023b) presents a comprehensive framework for empowering FNO to effectively handle non-uniformly distributed data. Xiao et al. (2024b) amortizes the parameters in FNO to accommodate arbitrarily many frequency modes to allow FNO to capture high-frequency details without having to have intractably many network parameters.

## 3 Background

### 3.1 Operator Learning

Operator learning refers to the use of neural networks to approximate an operator between infinite dimensional function spaces (Kovachki et al., 2023). Neural operators hold significant relevance in scientific computing, especially for solving PDEs. Concretely, consider a parametric PDE of the form:

$$\mathcal{N}(a, u)(x) = f(x), \quad x \in \Omega$$
$$u(x) = 0, \qquad x \in \partial\Omega$$

(2)

where $\Omega \subset \mathbb{R}^d$ is a bounded open set referred to as the domain of the PDE, $\mathcal{N}$ is a, possibly non-linear, differential operator, $a$ is the parametric input function, $f$ is a given fixed function in an appropriate function space determined by the structure of $\mathcal{N}$, and $u$ is the PDE solution we are interested in. The PDE solution operator is defined as $G(a) = u$. We are interested in learning the operator $G(\cdot)$ through a given finite collection of observations of input-output pairs $\{a_j, u_j\}_{j=1}^N$, where each $a_j$ and $u_j$ are functions provided in a discretized form.

### 3.2 Fourier Neural Operators

Let $\mathcal{N}$ in equation 2 be a linear differential operator, under fairly general conditions on $\mathcal{N}$, the solution operator can be defined as the following kernel integral operator:

$$u(x) = G(a) = \int_\Omega K_a(x, y) f(y) dy,$$

(3)

where $K_a$ is the kernel function defined on $\Omega \times \Omega$ and depends on the parameter function $a$. Drawing inspiration from the kernel integral method for the solution operator of PDEs, an integral neural operator comprises multiple layers of kernel integral operations with nonlinear activation functions to learn the solution

operator of diverse PDEs. Each layer is characterized by a fixed non-linearity and a fixed kernel integral operator $\mathcal{K}$ parameterized by network parameters, defined as

$$(\mathcal{K}v)(x) = \int \kappa(x, y)v(y)\mathrm{d}y, \tag{4}$$

where $v(x)$ is the input function to this kernel integral layer. FNO additionally enforces translation invariance on the kernel, expressed as $\kappa(x, y) = \kappa(x-y)$, which is a choice from the perspective of fundamental solutions. Hence, the kernel integral operator transforms into a convolution operator. Each Fourier layer in FNO is now characterized by a fixed non-linearity and a fixed convolution operator $\mathcal{K}$ parameterized by network parameters:

$$(\mathcal{K}v)(x) = \int_{\mathbb{R}^d} \kappa(x - y)v(y)dy. \tag{5}$$

Convolution can be efficiently computed as element-wise multiplication in the frequency:

$$(\mathcal{K}v)(x) = \mathcal{F}^{-1}(\mathcal{F}\kappa \cdot \mathcal{F}v)(x), \tag{6}$$

where $\mathcal{F}$ and $\mathcal{F}^{-1}$ are the Fourier transform and its inverse, respectively. FNO directly learns $\mathcal{F}\kappa$ in the Fourier domain as complex weights instead of learning the kernel $\kappa$ in the time space.

## 4    Dynamic Schwartz Fourier Neural Operator

This section introduces DSFNOs, a dynamic kernel adaptation of FNOs. DSFNOs feature two major modifications compared to FNOs: Firstly, the kernel is designed to be dynamic. Secondly, Schwartz operators are proposed to facilitate interactions between different frequency modes. By incorporating these two major components, our model gains the ability to discern relevant frequency information patterns, facilitating a better understanding and representation of complex physical phenomena. We start this section by introducing the Neural Kernel Generator (NKG) and the overall DSFNO framework in Section 4.1 and then we will define frequency domain interactions (FDIs), provide arguments for the need of FDIs, and discuss translation symmetry related to FDIs in Section 4.2. Lastly, based on the need for FDIs, we provide the formulation of Schwartz operators for generating dynamic kernels in Section 4.3.

### 4.1    Neural Kernel Generator

Kernel integral methods involve expressing the solution of a PDE as a convolution or integral involving a kernel function that represents the influence of one point in space on another. For example, in the case of elliptic equations, the solution of a linear, homogeneous PDE with a given forcing term can be expressed as a convolution of the Green's function with the forcing term. The readers are referred to Evans (2010) for more details on the kernel method for PDEs. In the kernel integral of the kernel method, namely equation 3, the kernel $K_a$ can depend on the parameter function $a$; however, in the kernel integral operator in FNO, namely equation 5, the kernel $\kappa$ is fixed. This can be justified as the parameter function will be the input to the neural operator in FNOs and Green's function method is just a motivation for the design of FNOs. However, this still motivates us to think that a dynamic kernel, that adjusts as the input changes, can enhance FNOs' expressiveness.

The core of DSFNO lies in the idea of utilizing dynamic kernels. In traditional convolution, either in FNOs or CNNs, static kernels are employed, where the weights remain constant throughout the operation once trained. However, dynamic kernels introduce a novel approach by leveraging a Neural Kernel Generator (NKG) to dynamically compute convolution kernels based on the input function: $\kappa_a = \kappa(a)$, where $a$ is the input function and $\phi$ is an operator that transforms the input into a convolution kernel function. We illustrate an FNO layer with dynamic kernel in Figure 2. In practice, the Neural Kernel Generator is a transformation mapping truncated Fourier coefficients of the input function to a band-limited output convolution kernel function. Such transformations can be as simple as a linear layer or as complex as a neural network such as the Spectral Neural Operator (Fanaskov & Oseledets, 2022b). It should be noted that NKG does not operate on infinite-dimensional spaces since we truncate the Fourier modes; however, NKGs are also resolution-invariant as they work with coefficients with respect to a function basis. Precisely, in a Fourier layer, the spectral convolution is now modified as follows:

$$(\mathcal{K}_a v)(x) = \int_{\mathbb{R}^d} \kappa(a)(x-y)v(y)dy, \tag{7}$$

which can also be efficiently computed as element-wise multiplication in the Fourier domain directly:

$$(\mathcal{K}_a v)(x) = \mathcal{F}^{-1}(\mathcal{F}\kappa(a) \cdot \mathcal{F}v)(x). \tag{8}$$

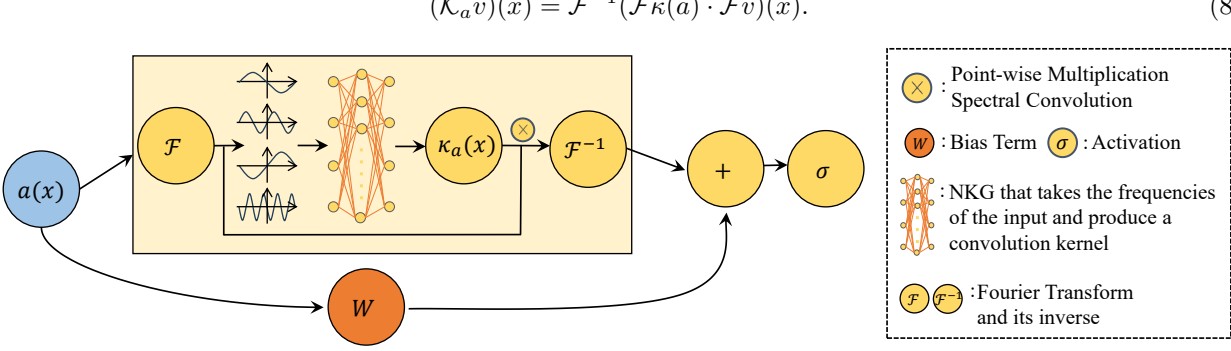

Figure 2: A Fourier layer with a Neural Kernel Generator (NKG): The input function $a(x)$ first goes through a Fourier transform to the frequency domain. Next, an NKG will take the frequencies of the input to generate a convolution kernel. Then, the input will be convoluted with the kernel through point-wise multiplication in the spectral domain. Subsequently, the result from the convolution will be inverse transformed back to the physical domain to include the bias term and undergo non-linear activation functions.

Additionally, the NKG can directly produce the coefficients of the kernel function in the Fourier domain, represented as $\mathcal{F}\kappa(a)$, instead of generating a function in the physical domain. Consequently, the output of the NKG can be seamlessly convoluted with the input function through element-wise multiplication in the Fourier domain, making the overall process highly efficient.

## 4.2 Frequency Domain Interaction through Neural Kernel Generator

Frequency Domain Interaction (FDI) refers to the way Fourier modes interact with each other. In FNO, in the spectral domain, there is no interaction between different Fourier modes since the resulting coefficients are obtained through point-wise multiplication with static kernels. In NKG, we allow connections between different modes to promote such interactions. We formally define frequency domain interactions as follows:

**Definition 4.1.** *Suppose $\mathcal{H}$ is a separable Hilbert space with an orthonormal basis $\{e_\ell\}_{\ell=0}^{\infty}$. We say that $T$ does not induce frequency domain interaction if for a function $v \in \mathcal{H}$, there exist fixed mappings $\{k_\ell : \mathbb{R} \mapsto \mathbb{R}\}_{\ell=0}^{\infty}$ with $\sum_{\ell=0}^{\infty} \|k_\ell(\cdot)\|_{max}^2 < \infty$ such that $\langle Tv, e_\ell \rangle = k_\ell(\langle v, e_\ell \rangle)$ for all $\ell$.*

In terms of FNO, the orthonormal basis in Definition 4.1 is the (generalized) Fourier basis. Essentially, FDI means that the update of one Fourier coefficient is dependent on the other Fourier coefficients. If we treat the truncated Fourier coefficients, denoted as $\hat{\mathbf{v}}$, as input to a linear layer $M\hat{\mathbf{v}}$ with a complex square matrix $M$, then there is no frequency domain interaction if and only if $M$ is diagonal. If we consider spectral convolution, i.e., point-wise multiplication in the spectral domain which can be thought of as $M$ being a square diagonal matrix, the update of one Fourier coefficient is not dependent on others. Without activation functions, an FNO layer consists of spectral convolution and possibly a residual connection; it is clear that such operations do not induce FDIs. An FNO layer includes activation functions to learn nonlinear behaviors and to recover the high frequency modes (Kovachki et al., 2023), which are truncated in the spectral convolution. We show that the nonlinear activation functions will also induce FDIs in order to learn nonlinear, diverse, and complex behaviors of physical systems.

**Theorem 4.2.** *Let $v(x)$ be a function in a separable Hilbert space such that $v(x)$ admits a Fourier transform, and let $\sigma(x)$ be a non-linear function. Assume that $\sigma(x)$ is analytic with respect to $x$. Then, applying the non-linear activation function $\sigma$ pointwise to $v(x)$, denoted mathematically as the function composition $\sigma \circ v(x)$, can induce interactions among all frequency modes.*

In pseudo-spectral methods, frequency domain interactions (FDIs) are common in the resulting linear systems (Gottlieb & Orszag, 1977). **As demonstrated in Theorem 4.2 (proof in Appendix B), nonlinear activation functions in FNO also induce FDIs. However, these FDIs are restricted to preserve translation equivariance, failing to allow the network to capture spatially varying dynamics effectively.** This restriction inspired us to incorporate unrestricted FDIs in the spectral domain, enhancing the ability of neural operators to learn complex and evolving behaviors of parametric PDEs. As illustrated in Figure 3, every Fourier mode is influenced by all others, supporting more intricate inter-frequency dynamics.

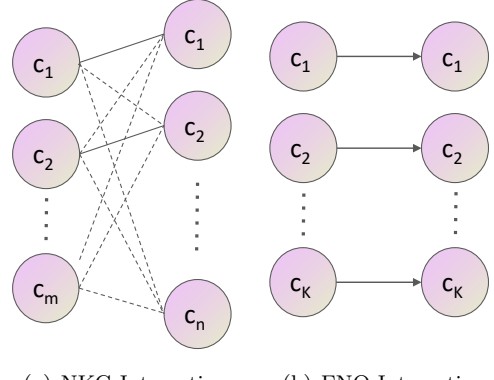

(a) NKG Interaction    (b) FNO Interaction

Figure 3: (a) The Neural Kernel Generator is not restricted to point-wise multiplication, allowing connections between different Fourier modes. The expressive power is enhanced through FDI. (b) An FNO Layer is based on global convolution, restricting the operations to point-wise multiplication in the frequency domain. Spectral convolution will not induce FDI in the spectral domain.

**Discussion on Translation Equivariance.** As FNO relies on convolution, it inherently maintains translational equivariance. However, within the DSFNO framework, where connections between different frequency modes are allowed in the neural kernel generator, this translational equivariance is no longer guaranteed. In Appendix C, we elaborate further on the basic concepts of translational equivariance and spectral convolutions. FNO can address this by employing padding or positional encoding (including grid coordinates as additional input channels) to remove incorrect translation symmetries, but restricting the integral operator solely to convolution integrals in FNO still limits the model's expressiveness. Through NKG, frequencies are naturally integrated; this integration removes unwanted translational symmetries in the model while also enhancing the learning capacities of FNOs. We provide a detailed justification for why FDIs in NKG can avoid the translational symmetries in Appendix C.2. As illustrated in Figure 4, imposing convolutions in general can limit the expressiveness of the model; even for operators with translational symmetries, it might still restrict the space of operators we can learn. *This might not be the case theoretically for FNO because FNO is a composition of linear integral operators and non-linear activation functions, and a translationally equivariant linear operator can always be written as a convolution-like integral, as a simple extension of Theorem 3.1 in Cohen et al. (2019).* However, practically, it might still limit the expressiveness of FNO. It will be of interest to theoretically investigate such aspects and explore how to design equivariant models for general neural operators, such as those in Fanaskov & Oseledets (2022b) and Lu et al. (2021).

### 4.3 Efficient Schwartz Kernel Operator

As illustrated in Section 4.2 , we aim to incorporate FDIs into NKGs. We can implement NKGs as a simple spectral neural operator (SNO) (Fanaskov & Oseledets, 2022a): we take truncated Fourier coefficients as input and produce the output coefficients with a fully connected neural network. Obviously, as the network is fully connected, it will induce FDIs. However, as we have multiple Fourier layers, the computational cost of using SNOs as NKGs can be prohibitive. Instead, we can effectively implement NKG simply as a linear operator. To formalize our approach, we introduce the Schwartz operator in the frequency domain, named in reference to the Schwartz kernel theorem, which asserts that

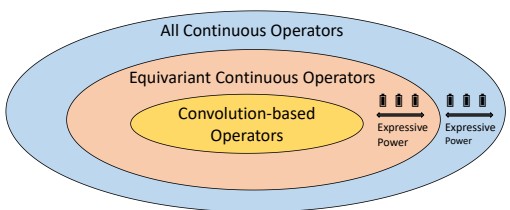

Figure 4: The paradigm of different neural operators. For example, DeepONet and SNO learn a general continuous operator while FNO learns an operator with translation equivariance.

**any linear operator can be represented as an integral over a kernel function**. With this foundation, we present Theorem 4.3, whose proof can be found in Appendix B, demonstrating that any integral

operator can be expressed as a frequency domain interaction. Essentially, the Schwartz kernel theorem allows us to represent any linear operator mapping between two functions as interactions in the frequency domain, given fairly general assumptions. As general integral kernels, Schwartz kernels are more expressive than convolution kernels in FNO, which is a special case of integral kernels.

**Theorem 4.3.** *An integral operator $T : H^s\left(\mathbb{R}^d; \mathbb{R}^{d_c}\right) \mapsto H^s\left(\mathbb{R}^d; \mathbb{R}^{d_c}\right)$ of the form $(Tv)(x) = \int_{\mathbb{R}^d} K(x, y)v(y)dy$, given that the kernel $K$ is smooth or sufficiently regular to ensure that its Fourier transform is well-behaved, can be expressed as*

$$(Tv)(x) = \mathcal{F}^{-1}\left(\int_{\mathbb{R}^d} M\left(\xi, \xi'\right) \hat{v}\left(\xi'\right) d\xi'\right), \tag{9}$$

*where $\hat{v}\left(\xi'\right)$ is the Fourier transform of $v(x)$, and $M\left(\xi, \xi'\right)$ is a complex function describing how the operator modulates the input function in the frequency domain.*

In this formulation, the Fourier transform of the output becomes a linear transformation of $\hat{v}(x)$ governed by $M(\xi, \xi')$. As a result, the coefficient value of any Fourier mode of the output is a weighted sum in the complex domain of all the Fourier modes of the input, and clearly, FDIs are induced through this process. In the case of the discrete Fourier transform, this corresponds to the $M\hat{v}$ formulation discussed earlier, without the need to impose a diagonal constraint on $M$. Furthermore, similar to the FNO, we can directly learn this complex matrix $M$ in the Fourier domain to promote efficiency. Although direct evaluation of this integral in practice involves $O(n^2)$ complexity, where $n$ is the size of the discretization of the functions. This complexity can be prohibitive, especially when compared to the more efficient $O(n \log n)$ complexity of the FNO. **However, a significant advantage of this Fourier space formulation lies in the global and compressive properties of Fourier transforms.** Similar to the truncation used in FNO, we can restrict the interaction to a subset of low-frequency modes, yielding a complexity of $O(k^2)$, where $k$ is the number of truncated Fourier modes. By doing so, we can still ensure global interactions in the physical space, effectively enhancing input interactions for a more expressive kernel. To further reduce computational complexity, albeit with some compromise in expressiveness, an efficient implementation can be found in Appendix D.

## 5 Experiments

In this section, we investigate whether DSFNO enhances the expressive power of FNO in terms of modeling complex dynamics more effectively. Different neural operators exhibit different advantages and demonstrate superior performance across different examples of PDEs; even within the same PDE, the performance can be influenced by the distribution of the data such as how the initial conditions are generated. While one neural operator may excel, being accurate and effective with a *significant* margin on one dataset, it might fail to perform well on another. There is no single state-of-the-art neural operator in the domain of neural PDE solvers. Given our primary objective to enhance FNO, we conduct experiments on two 2D examples from Li et al. (2021): the Incompressible Navier-Stokes equation and the Darcy flow equation. Additionally, we compare DSFNO with FNO on the 2D Shallow Water equation sourced from PDEbench (Takamoto et al., 2023). We also conduct an ablation study to systematically analyze the roles of the two components in DSFNO on the Navier-Stokes Equation.

**Baselines.** We include several meaningful baselines for comparison. First, we compare with *(1) FNO* (Li et al., 2021), as our proposed DSFNO framework aims to enhance the expressiveness of FNO, which serves as the backbone for many frameworks in diverse applications. *(2) FFNO* (Tran et al., 2023) is also included as it represents a refined version of FNO with separable spectral layers and improved residual connections. As FFNO has fewer learnable parameters, we compare with FFNO using the same channel width and modes, denoted as FFNO$^{\text{same}}$, as well as with a version using more modes and a larger channel width, simply denoted as FFNO. We test against *(3) CNO*, a key model based on fixed-size local convolutions, as well as *(4) DeepONet*, an influential model that employs learnable basis functions. For the Darcy flow example, we include two transformer-based models, specifically *(5) Galerkin Transformer (GT)* (Cao, 2021) and *(6) ONO* (Xiao et al., 2024a), though they are excluded from other examples due to poor performance, likely due to transformers' large data requirements or inefficiency to learn in an auto-regressive manner (auto-regressive settings are not implemented in the experiments from Xiao et al. (2024a); Cao (2021)). Notably, there has been a recent surge in transformer-based architectures. However, our main focus is on improving the

expressive power of FNO, and a comparison with all transformer-based architectures is not within the scope of this work. We provide further discussion on transformer-based models in Appendix E. Additionally, we include *(7) U-Net* from Takamoto et al. (2023) and *(8) GNO* (Anandkumar et al., 2019) for a comprehensive comparison. Many operators, such as DeepONet (Lu et al., 2021) and GNO (Anandkumar et al., 2019), are not configured to run in an autoregressive manner in their original setups. Therefore, for reference, we include the results of DeepONet (Lu et al., 2021) and GNO (Anandkumar et al., 2019) in directly predicting the final solution at the last time step given the initial time step for the Navier Stokes and shallow water equations, as we found that autoregressive results for these models fall significantly below expectations. This approach favors non-autoregressive models, as autoregressive models tend to suffer from error accumulation over time (Lippe et al., 2023). As highlighted in Lanthaler et al. (2024); Gao et al. (2025), neural operators can be significantly affected by discretization mismatch errors. Therefore, we do not conduct super-resolution tests or include super-resolution performance as an evaluation metric.

## 5.1 Incompressible Navier-Stokes Equation

**Simulation Description.** The Navier-Stokes equations stand as the cornerstone of fluid mechanics, governing the behavior of viscous fluids under various conditions. We consider the 2D Navier-Stokes equation for a viscous, incompressible fluid in vorticity form from Li et al. (2021):

$$\partial_t w(x,t) + u(x,t) \cdot \nabla w(x,t) = \nu \Delta w(x,t) + f(x),$$
$$\nabla \cdot u(x,t) = 0,$$
$$w(x,0) = w_0(x),$$

where $x \in \Omega = (0,1)^2$ is the spatial variable, $t \in \mathbb{T} = (0,T]$ is the temporal variable, $u \in C\left([0,T]; H^r_{\mathrm{per}}\left((0,1)^2; \mathbb{R}^2\right)\right)$ for any $r > 0$ is the velocity field, $w = \nabla \times u$ is the vorticity, $w_0 \in L^2_{\mathrm{per}}\left((0,1)^2; \mathbb{R}\right)$ is the initial vorticity, $\nu = 1e-3$ or $1e-4$ is the viscosity coefficient, and $f \in L^2_{\mathrm{per}}\left((0,1)^2; \mathbb{R}\right)$ is the forcing function. We are interested in learning the operator mapping the vorticity up to time 10 to the vorticity up to a later time $T = 50$ for $\nu = 1e-3$ and $T = 30$ for $\nu = 1e-4$ in an auto-regressive manner as in Li et al. (2021).

**Setup.** We follow the exact same setup as in Li et al. (2021) and directly present their autoregressive baseline results for reference including U-Net, TF-Net, and ResNet. For FNO, we report both the original results and the updated results from our experiments, reflecting modifications made by the authors to improve FNO's performance after the paper's publication. Further details about these experiments and the dataset are provided in Appendix F.

Table 1: Results on the Incompressible Navier-Stokes Equation: The mean and standard deviation of the average relative $\ell_2$ error are reported. The reported inference times are for one time step inference.

| | Common Information | | $\nu = 1e-3$ | $\nu = 1e-4$ |
|---|---|---|---|---|
| | # Parameters (M) | Inference Time (ms) | Test (%) | Test (%) |
| DSFNO | 1.06 | 3.05 | **0.562** $\pm$ 0.039 | **5.973** $\pm$ 0.286 |
| FNO | 0.93 | 3.01 | 0.795 $\pm$ 0.071 | 7.839 $\pm$ 0.176 |
| FFNO$^{\mathrm{same}}$ | 0.09 | 3.39 | 1.981 $\pm$ 0.068 | 10.523 $\pm$ 0.262 |
| FFNO | 0.44 | 6.16 | 1.073 $\pm$ 0.025 | 6.880 $\pm$ 0.105 |
| CNO | 2.66 | 20.54 | 1.957 $\pm$ 0.126 | 10.858 $\pm$ 0.239 |
| GNO $^{\diamond}$ | 4.61 | 124.62 | 26.973 $\pm$ 3.273 | 53.073 $\pm$ 0.926 |
| FNO$^{*}$ | 0.41 | N/A | 1.28 | 8.34 |
| U-Net$^{*}$ | 24.95 | N/A | 2.45 | 11.90 |
| TF-Net$^{*}$ | 7.45 | N/A | 2.25 | 11.68 |
| ResNet$^{*}$ | 0.26 | N/A | 7.01 | 23.11 |

[*]: Original error reported in Li et al. (2021), the FNO architecture had been updated after the publication. We report results on the new architecture provided in their repository for the fairness of comparison.

[$\diamond$]: Models that are not implemented in an auto-regressive manner.

[same]: FFNO reduces the number of parameters but does not improve runtime, as it is dominated by the FFT. Therefore, we conduct experiments using the same channel width and number of modes to ensure comparable inference times.

Table 2: Results on the Shallow Water Equation. The mean and standard deviation of the average relative $\ell_2$ error are reported. We observe that DSFNO can improve the expressiveness of FNO and achieve the best performance among all baselines.

| | #Par.(M) | Infer(ms) | Test (%) |
|---|---|---|---|
| DSFNO | 1.97 | 3.24 | **0.159** $\pm$ 0.015 |
| FNO | 1.54 | 3.02 | 0.216 $\pm$ 0.006 |
| FFNO$^{\text{same}}$ | 11.2 | 3.43 | 0.854 $\pm$ 0.058 |
| FFNO | 0.54 | 6.78 | 0.344 $\pm$ 0.012 |
| CNO | 2.66 | 20.37 | 0.269 $\pm$ 0.069 |
| U-Net | 7.76 | 2.67 | 3.964 $\pm$ 0.109 |
| GNO$^\diamond$ | 4.61 | 127.93 | 10.312 $\pm$ 0.952 |
| DeepONet$^\diamond$ | 2.40 | 1.63 | 1.070 $\pm$ 0.136 |

$^\diamond$ Models that are not implemented auto-regressively.

**Results and Analysis.** In Table 1, we present the results for the 2D Navier-Stokes equation. DSFNO outperforms all other baselines in terms of accuracy, which indicates that we can enhance the expressiveness of FNO through making the kernel dynamic with the proposed Schwartz operator for FDIs. It is observed that DSFNO incurs a similar inference time as FNO, this is because we implement an efficient version of NKG through Schwartz kernels, and the cost is mostly induced by the Fourier transform and its inverses. Additionally, the performance of DeepONet is particularly bad on this PDE example, with an error of over 20%; similar phenomena are observed in other works, such as Raonic et al. (2023). However, we want to point out that DeepONet is effective and accurate on many PDE examples (Lu et al., 2022) and DeepONet can work effectively and efficiently on non-rectangular domains as well; some additional architectural and training changes might significantly improve the performance of DeepONet (Lu et al., 2022), although none of our attempts are successful. Therefore, to not misguide the readers and leave an impression that DeepONet is suboptimal, we decide to not include its results in Table 1. We provide the results in Appendix F.1.1. Additionally, we provide results on the memory and runtime comparison between DSFNO and FNO in Appendix F.1.1.

## 5.2 The Shallow Water Equation

**Simulation Description.** We consider the following form of a system of hyperbolic PDEs from Takamoto et al. (2023):

$$\partial_t h + \partial_x hu + \partial_y hv = 0,$$

$$\partial_t hu + \partial_x \left( u^2 h + \frac{1}{2}g_r h^2 \right) = -g_r h \partial_x b, \tag{10}$$

$$\partial_t hv + \partial_y \left( v^2 h + \frac{1}{2}g_r h^2 \right) = -g_r h \partial_y b,$$

with a spatial domain $\Omega = [-2.5, 2.5]^2$, $u, v$ being the velocities in horizontal and vertical directions, $h$ describing the water depth and $b$ describing a spatially varying bathymetry. $hu, hv$ can be interpreted as the directional momentum components and $g_r$ describes the gravitational acceleration. We are interested in learning the operator mapping the initial condition $h(t = 0, x, y)$ to a later time $h(t = T, x, y)$.

**Setup and Results.** We use the dataset provided by Takamoto et al. (2023), details of which can be found in the Appendix F.1.3. Originally, the data is given in a spatial resolution of $128 \times 128$ and a temporal resolution of 101; we downsampled this data to a spatial resolution of $64 \times 64$ and a temporal resolution of 25. The shallow water equation is time-dependent, all models except GNO and DeepONet are trained in an autoregressive manner using the first 5 time steps as inputs and predicting the subsequent 20 time steps.

In Table 3, we present the results for the 2D shallow water equation. Similar to the Navier-Stokes example, DSFNO outperforms all baselines in terms of accuracy. It suggests that DSFNO enhances the expressiveness of FNO. Similarly, we do not observe any significant increase in computational costs.

### 5.3 The Darcy Flow Equation

**Simulation Description.** We consider the steady-state of the 2D Darcy Flow equation from Li et al. (2021) given by:

$$-\nabla \cdot (a(x)\nabla u(x)) = f(x), \quad x \in (0,1)^2$$
$$u(x) = 0, \qquad x \in \partial(0,1)^2$$

where $a \in L^\infty\left((0,1)^2; \mathbb{R}_+\right)$ is the diffusion coefficient and $f \in L^2\left((0,1)^2; \mathbb{R}\right)$ is the forcing function that is kept fixed $f(x) = 1$. We are interested in learning the operator mapping the diffusion coefficient $a(x)$ to the solution $u(x)$.

**Setup and Results.** We use the data generator provided by Li et al. (2021). The input diffusion coefficient field $a(x,y)$ is generated by a Gaussian random field with a piecewise function. Details of the data generation can be found in Appendix F.1.2. The Darcy flow equation we consider does not exhibit translation symmetries, for convolution-based models, we apply padding and concatenate coordinate grids to avoid unintended translational symmetry in the model. As the input size of the Darcy flow equation is not a power of 2, for U-Net, we apply additional padding to the transposed convolutional layers to ensure correct output sizes. Alternatively, we can apply padding or interpolation to the input to obtain inputs of size $64 \times 64$, but these options result in slightly worse performance. Therefore, we opt to apply padding in the up-convolution layers.

In Table 3, we present the results for the 2D Darcy flow equation. Similar to what we observe in the Navier-Stokes example, DSFNO outperforms all baselines in terms of accuracy. It suggests that DSFNO enhances the expressiveness of FNO. Similarly, we do not observe any significant increase in computational costs.

Table 3: Results on the Darcy Flow Equation. The mean and standard deviation of the relative $\ell_2$ error are reported. We observe that DSFNO can improve the expressiveness of FNO and achieve the best performance among all baselines.

| | #Par.(M) | Infer(ms) | Test (%) |
|---|---|---|---|
| DSFNO | 2.77 | 2.87 | **0.659** $\pm$ 0.005 |
| FNO | 2.36 | 2.22 | 0.788 $\pm$ 0.005 |
| FFNO$^{\text{same}}$ | 0.27 | 2.73 | 0.754 $\pm$ 0.009 |
| FFNO | 1.72 | 3.92 | 0.676 $\pm$ 0.023 |
| GT | 1.99 | 9.95 | 0.913 $\pm$ 0.076 |
| ONO | 1.01 | 153.83 | 1.171 $\pm$ 0.143 |
| CNO | 2.61 | 17.83 | 0.762 $\pm$ 0.014 |
| U–Net | 7.76 | 2.33 | 1.294 $\pm$ 0.003 |
| GNO | 4.61 | 98.72 | 3.392 $\pm$ 0.981 |
| DeepONet | 2.48 | 1.91 | 2.987 $\pm$ 0.099 |

Table 4: Results of the ablation study. The mean and standard deviation of the average relative $\ell_2$ error are reported. Clearly, both adaptive kernels and frequency domain interactions can enhance the performance of FNO. However, DSFNO significantly improves performance by a larger margin, indicating that both components are indispensable for designing an expressive DSFNO.

| Model | Test (%) |
|---|---|
| DSFNO | **0.562** $\pm$ 0.039 |
| FNO | 0.795 $\pm$ 0.071 |
| Adaptive Kernel Only | 0.679 $\pm$ 0.029 |
| FDI Only | 0.704 $\pm$ 0.027 |

### 5.4 Ablation Study on the Navier Stokes Equation

We perform an ablation study to assess the separate contributions of kernel adaptivity and frequency domain interactions (FDI) within DSFNO. We systematically analyze two DSFNO variants: one with only the dynamic kernel mechanism (no FDI) and the other with only FDI (no dynamic kernels). In the first variant, as defined in Definition 4.1, the NKG is limited to transformations within individual frequency modes, such as a pointwise neural network or simple pointwise multiplication with learnable weights. This study isolates the effects of each component on model performance, clarifying the impact of dynamic kernels and FDI on predictive accuracy in modeling complex operators.

We present the results in Table 4, both variants,dynamic kernel-only and FDI-only, demonstrate slightly better performance compared to the original FNO and barely increase the computation time. This indicates that

each component independently contributes to enhancing FNO's effectiveness and expressiveness. Moreover, DSFNO, which integrates both dynamic kernel and FDI, consistently outperforms both individual variants. This comprehensive approach leverages synergies between dynamic kernels and frequency interactions, resulting in the highest observed performance improvements. These findings underscore the significance of incorporating both dynamic kernel mechanisms and FDI to optimize and advance the capabilities of neural operators in solving complex problems efficiently.

## 6 Conclusion and Future Work

In this work, we propose to design a DSFNO architecture based on dynamic kernels. Specifically, by considering a dynamic kernel that is adaptive to the input and also induces frequency domain interactions, DSFNO gains the ability to perceive relevant frequency information patterns, facilitating better understanding and representation of complex physical phenomena. We conduct experiments to evaluate our proposed DSFNO. Results show that Fourier neural operators with a dynamic kernel can achieve better performance under similar settings and computational budgets. These results underscore the effectiveness of our proposed approach in using dynamic kernels in Fourier neural operators. Overall, our findings underscore the potential of dynamic kernels in advancing the capabilities of Fourier neural operators, providing a more robust framework for tackling intricate problems in various scientific and engineering domains.

**Future Work.** It will be interesting to see if designing a more sophisticated neural kernel generator could further improve DSFNO and yield other significant benefits, such as enabling learning of more complex dynamics, reducing the latent dimension (channel dimension) of FNO, and speeding up both training and inference by minimizing FFT costs associated with high channel dimensions. Furthermore, as noted in the discussion on translation symmetries in Section 4.2, there is a compelling interest in developing general equivariant methods tailored for other operators such as SNO (Fanaskov & Oseledets, 2022a), and DeepONet (Lu et al., 2021). Additionally, it will be interesting to see if DSFNO can serve as a better alternative to FNO in learning foundation models for operator learning (Subramanian et al., 2023).

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

# A    Additional Visualization of Learned Kernels

We provide additional visualization of the learned kernels of DSFNO in Fig. 5

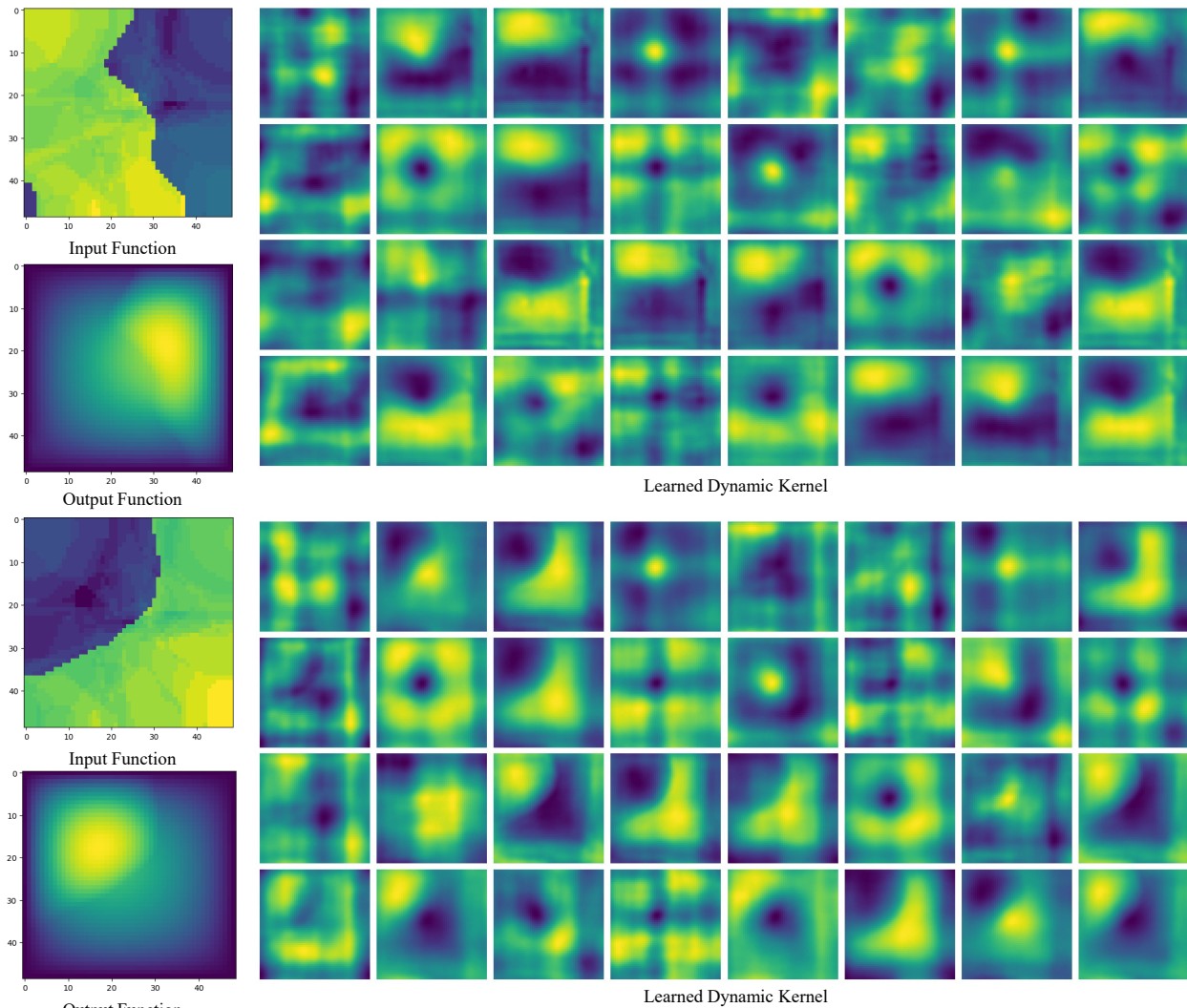

Figure 5: Visualization of two different pairs of input and output functions on the Darcy flow PDE with their corresponding learned kernels. Note that there are $L \times C_{in} \times C_{out}$ DSFNO kernels in total; we present the dynamic kernels for the in-channels in the first layer.

# B    Proofs

## B.1    Proof of Theorem 4.2

**Theorem 4.2.** *Let $v(x)$ be a function in a separable Hilbert space such that $v(x)$ admits a Fourier transform, and let $\sigma(x)$ be a non-linear function. Assume that $\sigma(x)$ is analytic with respect to $x$. Then, applying the non-linear activation function $\sigma$ pointwise to $v(x)$, denoted mathematically as the function composition $\sigma \circ v(x)$, can induce interactions among all frequency modes.*

*Proof.* Let $\hat{v}(\xi) = \mathcal{F}[v(x)](\xi)$ denote the Fourier transform of $v(x)$ given by $\hat{v}(\xi) = \int_\Omega v(x)e^{-2\pi i\xi x}dx$. If $\sigma(x)$ is a smooth function of $x$, then we can expand $\sigma$ as

$$\sigma(x) = \sigma(0) + \sigma'(0)x + \frac{1}{2!}\sigma''(0)x^2 + \frac{1}{3!}\sigma'''(0)x^3 + \dots \tag{11}$$

Then,

$$\sigma \circ v(x) = \sigma(0) + \sigma'(0)v(x) + \frac{1}{2!}\sigma''(0)v(x)^2 + \dots \tag{12}$$

and we perform the Fourier transformation,

$$\mathcal{F}[\sigma \circ v(x)](\xi) = 2\pi\sigma(0)v(\xi) + \sigma'(0)\hat{v}(\xi) + \frac{1}{2!}\sigma''(0)\int_\Omega \frac{d\xi_0}{2\pi}\hat{v}(\xi_0)\,\hat{v}(\xi - \xi_0)\,d\xi_0$$

$$+ \frac{1}{3!}\sigma'''(0)\iint_\Omega \frac{d\xi_0}{2\pi}\frac{d\xi_1}{2\pi}\hat{v}(\xi_0)\,\hat{v}(\xi_1 - \xi_0)\,\hat{v}(\xi - \xi_1)\,d\xi_0 d\xi_1 + \dots.$$

Here, we observe that, without loss of generality, as long as there exists $n > 1$, and $\sigma^n(0)$ is non-zero at 0, such function composition induces interactions among all Fourier modes. Practically, most non-linear functions satisfy these constraints. In practice, we empirically observe that, even non-linear functions that do not satisfy these constraints, such as ReLU, will still promote interactions among Fourier modes. $\square$

## B.2 Proof of Theorem 4.3

**Theorem 4.3.** *An integral operator* $T : H^s\left(\mathbb{R}^d; \mathbb{R}^{d_c}\right) \mapsto H^s\left(\mathbb{R}^d; \mathbb{R}^{d_c}\right)$ *of the form* $(Tv)(x) = \int_{\mathbb{R}^d} K(x,y)v(y)dy$, *given that the kernel* $K$ *is smooth or sufficiently regular to ensure that its Fourier transform is well-behaved, can be expressed as*

$$(Tv)(x) = \mathcal{F}^{-1}\left(\int_{\mathbb{R}^d} M(\xi, \xi')\,\hat{v}(\xi')\,d\xi'\right), \tag{9}$$

*where* $\hat{v}(\xi')$ *is the Fourier transform of* $v(x)$, *and* $M(\xi, \xi')$ *is a complex function describing how the operator modulates the input function in the frequency domain.*

*Proof.* First of all, given the assumption and Sobolev function spaces, it is safe to conclude that all the functions' Fourier transforms are well-behaved. Take the Fourier transform of both sides of the equation

$$\mathcal{F}\{(Tv)(x)\} = \mathcal{F}\left\{\int_{\mathbb{R}^d} K(x,y)v(y)dy\right\}$$

$$= \int_{\mathbb{R}^d}\left(\int_{\mathbb{R}^d} K(x,y)v(y)dy\right)e^{-i\xi x}dx$$

$$= \int_{\mathbb{R}^d} v(y)\left(\int_{\mathbb{R}^d} K(x,y)e^{-i\xi x}dx\right)dy$$

$$= \int_{\mathbb{R}^d} \hat{K}(\xi, y)v(y)dy$$

$$= \int_{\mathbb{R}^d} \hat{K}(\xi, y)\left(\int_{\mathbb{R}^d} \hat{v}(\xi')\,e^{i\xi' y}d\xi'\right)dy.$$

Define $\hat{K}(\xi, y) = \int_{\mathbb{R}^d} K(x,y)e^{-i\xi x}dx$ as the Fourier transform of $K$ with respect to $x$, we have

$$\mathcal{F}\{(Tv)(x)\} = \int_{\mathbb{R}^d} \hat{K}(\xi, y)v(y)dy$$

$$= \int_{\mathbb{R}^d} \hat{K}(\xi, y)\left(\int_{\mathbb{R}^d} \hat{v}(\xi')\,e^{i\xi' y}d\xi'\right)dy,$$

$$= \int_{\mathbb{R}^d} \hat{v}(\xi')\left(\int_{\mathbb{R}^d} \hat{K}(\xi, y)e^{i\xi' y}dy\right)d\xi'$$

$$= \int_{\mathbb{R}^d} \hat{v}(\xi')\,M(\xi, \xi')\,d\xi'$$

where $v(y) = \int_{\mathbb{R}^d} \hat{v}(\xi') e^{i\xi'y} d\xi'$ is the Fourier transform of $v$ and $M(\xi, \xi') = \int_{\mathbb{R}^d} \hat{K}(\xi, y) e^{i\xi'y} dy$ is the Fourier transform of $\hat{K}(\xi, y)$ with respective to $y$; in other words, $M(\xi, \xi')$ is the Fourier transform of $K(x, y)$ over both $x$ and $y$. This completes the proof by applying the inverse Fourier Transform to both sides. $\square$

## C    Preliminaries

### C.1    Group Equivariance and Translation Symmetries in FNO

Equivariance is a property of an operator, such as the solution operator of PDEs, a neural operator, or a layer within a neural operator, such that if the input transforms, the output transforms correspondingly in a predictable way.

**Definition C.1** (Equivariance). *An operator $\Phi : \mathcal{X} \to \mathcal{Y}$ is said to be equivariant to a group $G$ if actions of $G$ on $\mathcal{X}$ and $\mathcal{Y}$, respectively denoted by $L_g^{\mathcal{X}} : \mathcal{X} \to \mathcal{X}$ and $L_g^{\mathcal{Y}} : \mathcal{Y} \to \mathcal{Y}$, satisfy*

$$\forall g \in G: \ L_g^{\mathcal{Y}}(\Phi(f)) = \Phi(L_g^{\mathcal{X}} f), \ f \in \mathcal{X},$$

*where $\mathcal{X}$ and $\mathcal{Y}$ are subspaces of a function space. In other words, the operator $\Phi$ commutes with actions of $G$.*

In the context of operator learning, $L_g^{\mathcal{X}}$ and $L_g^{\mathcal{Y}}$ can be considered as transformations, such as translation or rotation, of the input parameter function and the output solution function, respectively. $\Phi$, on the other hand, can be thought of as the solution operator that maps an input function to its corresponding output solution.

Equivariance to various groups, including but not limited to $SE(n), E(n), \mathrm{SIM}(n)$, can be achieved by group-convolutions (Cohen & Welling, 2016; Weiler et al., 2023) or group/frame averaging (Puny et al., 2022).

Let $v(h)$ and $\kappa(h)$ be real-valued functions on group $G$ with $L_g v(h) = v\left(g^{-1}h\right)$, the group convolution is defined as:

$$(v \star \kappa)(h) = \int_{g \in G} v(g)\kappa\left(g^{-1}h\right) dg,$$

where $v$ can be regarded as the input (feature) function and $\kappa$ the convolution kernel.

Integrability over a group and the identification of the suitable measure, $dg$, are necessary for group convolution. It has been shown that with the measure $dg$, group convolution consistently maintains group equivariance. For all $a \in G$,

$$\begin{aligned}
(L_a v \star \kappa)(h) &= \int_{g \in G} v\left(a^{-1}g\right) \kappa\left(g^{-1}h\right) dg \\
&= \int_{b \in G} v(b)\kappa\left((ab)^{-1}h\right) db \\
&= \int_{b \in G} v(b)\kappa\left(b^{-1}a^{-1}h\right) db \\
&= (v \star \kappa)\left(a^{-1}h\right) \\
&= L_a (v \star \kappa)(h).
\end{aligned}$$

Each Fourier layer in FNO consists of a point-wise nonlinear activation function and a convolution over the domain $\Omega \subset \mathbb{R}^d$:

$$(\mathcal{K}v)(x) = \int_{\mathbb{R}^d} \kappa(x - y)v(y) dy, \tag{13}$$

it follows immediately that Fourier layers are equivariant to the translation group $\mathbb{R}^d$.

### C.2    Frequency Domain Interaction Breaks Translation Equivariance

While the Fourier Neural Operator (FNO) is equivariant to translation, the frequency domain interactions (FDIs) can break this equivariance. At a high level, the simplest FDIs can be seen as a weighted summation

in the frequency domain. As shown in Appendix B.2, a weighted summation in the frequency domain is no longer a convolution but a full integral $\int_\Omega k\left(x, y\right) v(y) dy$ in the real space. Thus, it is naturally not equivariant. This can also be seen purely from a frequency domain perspective. A translation in the input function corresponds to a phase shift in the Fourier domain:

$$v\left(x\right) \xrightarrow{\mathcal{F}} e^{-i\xi}\hat{v}(\xi) \quad \text{and} \quad v\left(x - x_0\right) \xrightarrow{\mathcal{F}} e^{-i\xi x_0}\hat{v}(\xi),$$

where $v(x)$ is the original function, $v(x-x_0)$ is the translated function by $x_0$, $\hat{v}(\xi)$ is the Fourier transform of $v(x)$, $\mathcal{F}$ denotes the Fourier transform operator, and $\xi$ is the frequency variable in the Fourier domain. The spectral convolution in FNO is a point-wise multiplication in the Fourier domain by the Fourier convolution theorem:

$$k(x) \star v\left(x\right) \xrightarrow{\mathcal{F}} e^{-i\xi}(\hat{k} \cdot \hat{v})(\xi) \quad \text{and} \quad k(x) \star v\left(x - x_0\right) \xrightarrow{\mathcal{F}} \hat{k}(\xi)e^{-i\xi x_0}\hat{v}(\xi) = e^{-i\xi x_0}(\hat{k} \cdot \hat{v})(\xi).$$

Clearly, FNO preserves translation equivariance because the phase shift $e^{-i\xi x_0}$ remains unaffected by the pointwise multiplication in the Fourier domain. However, if we introduce FDIs, the phase shift property is disrupted. Suppose a general integral transformation $\mathcal{T}$ is applied to the Fourier domain to have FDIs:

$$\mathcal{T}(\hat{v}(\xi)) = \int w\left(\xi, \xi'\right)\hat{v}\left(\xi'\right) d\xi'$$

where $w\left(\xi, \xi'\right)$ represents the interaction weights across frequencies. Applying this transformation to the translated function:

$$\mathcal{T}\left(e^{-i\xi x_0}\hat{v}(\xi)\right) = \int w\left(\xi, \xi'\right) e^{-i\xi' x_0}\hat{v}\left(\xi'\right) d\xi'$$

Unlike the case of pointwise multiplication, the phase shift $e^{-i\xi x_0}$ is no longer preserved consistently across frequencies due to the integral over $\xi'$. Consequently, the transformation distorts the structured phase relationship, breaking translation equivariance.

## D   Efficient Implementation of Frequency Mode Interaction

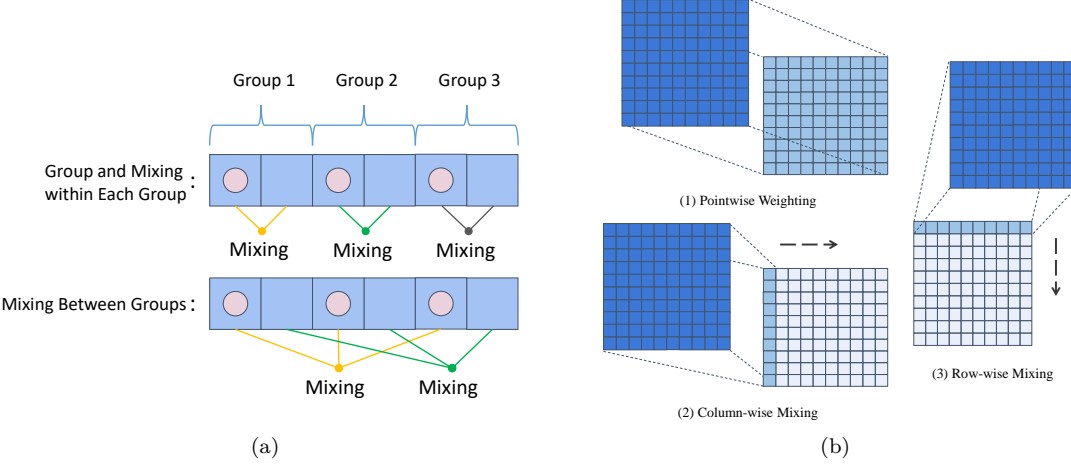

Figure 6: (a) Efficient Frequency Domain Interaction by Group and Mixing: First of all, we group the frequencies and perform mixing within each group. Secondly, we perform mixing between different groups. (b) Efficient Linear NKG in 2D: For a 2D array of frequencies (Fourier coefficients), we first apply a kernel point-wise to the frequencies; this is to learn a general pattern for individual frequencies. Then, we perform mixing for each row and column, separately. Row-wise mixing combined with column-wise mixing learns the interaction across all frequency modes.

As illustrated in Sec. 4.1 and Sec. 4.2, we aim to incorporate FDIs into NKGs. We can implement NKGs as a simple spectral neural operator (SNO) (Fanaskov & Oseledets, 2022a): we take truncated Fourier coefficients as input and produce the output coefficients with a fully connected neural network. Obviously, as the network is fully connected, it will induce FDIs. However, as we have multiple Fourier layers, the computational cost of using SNOs as NKGs can be prohibitive. Instead, we implement NKG simply as a linear layer operating on coefficients: we take the truncated Fourier coefficients, denoted as $\hat{\mathbf{v}}$, as input to a linear layer $M\hat{\mathbf{v}}$. Here, we no longer restrict $M$ to be square diagonal as in spectral convolution. It is obvious that such linear layer will induce FDIs.

Practically, let's assume we have real-valued discrete signals of resolution $N$ (with $N$ equidistant grid points in the domain). The Fast Fourier Transform (FFT) incurs a complexity of $O(N \log N)$. Usually, we truncate the number of Fourier modes to some $k < N$ to perform convolution through the spectral domain, thus the complexity of a linear NKG is $O(k^2)$. For a relatively large $k$, which can go as high as $N$, $O(k^2)$ can still result in significant computational costs.

We propose an efficient FDI grouping method. For the $k$ truncated frequency terms, we group them into $m$ groups, with each group containing $\frac{k}{m}$ frequency terms, assuming $m$ divides $k$. A weight matrix $W_1$ of size $\frac{k}{m} \times \frac{k}{m}$ will be applied to each group to perform mixing within the group. We perform mixing between different groups by mixing elements from all groups. This grouping and mixing process is illustrated in part (a) of Figure 6. Although this grouping may seem complicated, it is natural to consider when the underlying physical system is in 2D or higher dimensions. We will illustrate this with a 2D example, and for higher dimensions, this mixing can even be done recursively to further reduce complexity.

Given the discrete equidistant sensor points on a 2D rectangular domain as a 2D array, the resulting Fourier frequencies will also be a 2D array, and we might truncate to only keep the low-frequency modes. Let $r \times c$ denote the size of the truncated frequency array. The efficient linear NKG is performed in three steps, as illustrated in part (b) of Figure 6: First, we apply a kernel point-wise to the frequencies; this is to learn a general pattern for individual frequencies. Second, we multiply a kernel of size $r \times r$ to every column of the resulting frequency array; this is to learn the interaction within every column. Lastly, we multiply a kernel of size $c \times c$ to every row of the resulting frequency array; this is to learn the interaction within every row. Row-wise mixing combined with column-wise mixing learns the interaction across all frequency modes. If the original truncated frequencies are of size $k \times k$, while the full matrix multiplication interaction would cost $O(k^4)$, the efficient Frequency Domain Interaction lowers the cost to $O(k^3)$. The number of parameters involved is also reduced from $O(k^4)$ to $O(k^2)$, significantly speeding up the training process and reducing memory usage as a smaller computational graph (for backward propagation) is constructed.

# E    Recent Transformer-based Models

For complex problems involving turbulent characteristics and small-scale features, such as the Navier-Stokes equations with a very high Reynolds number, global operations are often prone to oversmoothing (Liu-Schiaffini et al., 2024). Recently, there has been a surge in transformer-based neural operators, which leverage the powerful self-attention mechanisms to capture long-range dependencies while maintaining flexibility in handling multi-scale features. GT (Cao, 2021) is the pioneering work that relates attention with a kernel integral in the field of operator learning. Many PDE systems are typically discretized into large-scale meshes with complex geometries, a lot of works have been proposed to mitigate such issues; for example, FactFormer (Li et al., 2023) factorizes the kernel integral to scale better with the dimensionality of the PDE problem, Transolver (Wu et al., 2024) splits the discretized domain into a series of learnable slices of flexible shapes, and LSM (Wu et al., 2023) removes the reliance on a coordinate space and utilizes an attention-based hierarchical projection network to efficiently map high-dimensional data into a compact latent space with linear-time complexity.

While transformer-based methods have shown great potential in solving high-dimensional multi-scale PDE problems, FNO and its variants remain prominent due to their efficiency in global feature extraction. DSFNO explores the expressive power of FNO and introduces a new adaptive kernel for improved performance. We believe that the most valuable comparison for DSFNO is against FNO. FNO and DSFNO tend to perform better than transformer-based models when global patterns are more prominent due to their ability to

efficiently and accurately capture such patterns. However, they can be prone to over-smoothing when small-scale features are more significant, whereas transformer-based models excel in such tasks. Such issues with FNO have been noted and can be mitigated in [1]. The same approach can be taken for our proposed DSFNO as well; however, it is beyond the scope of our work. In summary, we believe that improving FNO is another important path towards advancing the field of operator learning and is just as crucial as transformer-based models. Both of these two mainstream approaches have their distinct advantages, and comparisons within each branch are more insightful.

## F   Implementation Details

### F.1   Data and Experimental Setups

#### F.1.1   The Navier-Stokes Equation

We consider the 2D Navier-Stokes equation for a viscous, incompressible fluid in vorticity form on the unit torus from Li et al. (2021):

$$\partial_t w(x,t) + u(x,t) \cdot \nabla w(x,t) = \nu \Delta w(x,t) + f(x), \ \ x \in \Omega, t \in \mathbb{T}$$
$$\nabla \cdot u(x,t) = 0, \qquad\qquad\qquad x \in \Omega, t \in \mathbb{T}$$
$$w(x,0) = w_0(x), \qquad\qquad\quad x \in \Omega$$

where $\Omega = (0,1)^2$ is the spatial domain, $\mathbb{T} = (0,T]$ is the temporal domain. The initial condition $w_0(x)$ is generated according to $w_0 \sim \mu$ where $\mu = \mathcal{N}\left(0, 7^{3/2}(-\Delta + 49I)^{-2.5}\right)$ with periodic boundary conditions. The forcing is kept fixed $f(x) = 0.1\left(\sin\left(2\pi\left(x_1 + x_2\right)\right) + \cos\left(2\pi\left(x_1 + x_2\right)\right)\right)$. The equation is solved using the stream-function formulation with a pseudospectral method. First, a Poisson equation is solved in Fourier space to find the velocity field. Then the vorticity is differentiated and the non-linear term is computed in physical space, after which it is dealiased. Time is advanced with a Crank-Nicolson update where the non-linear term does not enter the implicit part. All data are generated on a $256 \times 256$ grid and are downsampled to $64 \times 64$. We use a time-step of $1e-4$ for the Crank-Nicolson scheme in the data-generated process where we record the solution every $t = 1$ time units.

**Setup.** We use the dataset provided by the authors of Li et al. (2021) on their GitHub repository and present their autoregressive baseline results, including U-Net, TF-Net, and ResNet, for reference directly. This choice is made because these models are not available in their repository and while we attempt to re-implement these models in an auto-regressive way, the resulting performance falls below what is presented in their paper. Moreover, many operators, such as DeepONet (Lu et al., 2021) and GNO (Anandkumar et al., 2019), do not run in an auto-regressive manner. While we attempt to re-implement these operator models with an auto-regressive approach, the resulting performance falls significantly below expectations, potentially because of high one-step errors that accumulate through time steps. This could be attributed to either the inherent characteristics of the models or the need for **substantial** additional engineering work. Therefore, for reference, we include the results on DeepONet (Lu et al., 2021) and GNO (Anandkumar et al., 2019) in predicting directly the final solution at the last time step given the initial time step. This would be in favor of non-autoregressive models as autoregressive models will face the issue of error accumulation through time (Lippe et al., 2023). Furthermore, we include updated results on FNO from our experiments, reflecting some modifications made to the model by the authors to improve FNO's results after the paper's publication. The average relative $\ell_2$ errors are reported. For auto-regressive models, $\ell_2$ errors reported are averaged over all time steps. All of our results are averaged over 10 runs for $\nu = 1e-3$ and averaged over 5 runs for $\nu = 1e-4$ due to the extensive computation requirements. Average inference time is measured on batches of size 20. As FNO and DKO are auto-regressive, we provide the inference time for only one time step for fair comparison with other non-auto-regressive models. Following the setups in Li et al. (2021), the training dataset contains 1000 and 10000 input-output function pairs for $\nu = 1e-3$ and $\nu = 1e-4$ respectively, whereas the testing dataset contains 200 input-output function pairs. The primary objective of our proposed DKO framework is to enhance the expressiveness of FNO, which is the backbone model of many frameworks for diverse applications. Our primary comparison baseline will be against FNO itself.

**Results of DeepONet.** The results of DeepONet on the Navier-Stokes equation are listed in Table 5. Again, we want to emphasize that DeepONet is an efficient and effective neural operator. Similar phenomena where DeepONet might observe suboptimal performance are presented in other works, such as Raonic et al. (2023). This should be investigated further in the AI4PDE community, although none of our attempts have been successful.

Table 5: Results on the Incompressible Navier-Stokes Equation.

|  | Common Information | | $\nu = 1e-3$ | $\nu = 1e-4$ |
|---|---|---|---|---|
|  | # Par. (M) | Inference Time (ms) | Test (%) | Test (%) |
| DeepONet$^\diamond$ | 2.40 | 1.63 | $18.865 \pm 5.143$ | $26.381 \pm 5.244$ |

$^\diamond$ Models that are not implemented in an auto-regressive manner.

**Training Time and Peak Memory Comparison between FNO and DSFNO.** We report the training time per epoch and peak GPU memory comparison between FNO and DSFNO on the Navier-Stokes equation with a viscosity constant of $1e-3$ in Table 6 to evaluate their computational efficiency. The training time per epoch is reported on *NVIDIA Tesla V100 Volta GPU Accelerator 32GB Graphics Card*. The standard deviation is several orders smaller than the mean, we only report the mean.

Table 6: Training Time and Peak Memory.

| Model | Training Time (s) | Peak Memory (MiB) |
|---|---|---|
| FNO | 18.76 | 12034 |
| DSFNO | 21.59 | 12238 |

### F.1.2 The Darcy Flow Equation

The Darcy flow equation is a fundamental equation in fluid dynamics that describes the flow of fluids through porous media. It is crucial in various fields like hydrogeology, petroleum engineering, soil mechanics, and environmental science. We consider the steady-state of the 2D Darcy Flow equation from Li et al. (2021) given by:

$$-\nabla \cdot (a(x)\nabla u(x)) = f(x), \quad x \in (0,1)^2$$
$$u(x) = 0, \qquad x \in \partial(0,1)^2$$

where $a \in L^\infty\left((0,1)^2; \mathbb{R}_+\right)$ is the diffusion coefficient and $f \in L^2\left((0,1)^2; \mathbb{R}\right)$ is the forcing function that is kept fixed $f(x) = 1$. We are interested in learning the operator mapping the diffusion coefficient $a(x)$ to the solution $u(x)$.

The input diffusion coefficient field $a(x,y)$ is generated by a Gaussian random field with a piecewise function, namely $a(x,y) = \psi(\mu)$, where $\mu$ is a distribution defined by $\mu = \mathcal{N}\left(0, (-\Delta + 9I)^{-2}\right)$. The mapping $\psi : \mathbb{R} \to \mathbb{R}$ takes the value 12 for positive values and 3 for negative values, and the push-forward is defined point-wise. Solutions are obtained using a second-order finite difference scheme on a $241 \times 241$ grid and then downsampled to $49 \times 49$.

### F.1.3 The Shallow Water Equation

We consider the following form of a system of hyperbolic PDEs from Takamoto et al. (2023):

$$\partial_t h + \partial_x hu + \partial_y hv = 0,$$

$$\partial_t hu + \partial_x \left(u^2 h + \frac{1}{2} g_r h^2\right) = -g_r h \partial_x b,$$

$$\partial_t hv + \partial_y \left(v^2 h + \frac{1}{2} g_r h^2\right) = -g_r h \partial_y b,$$

with a spatial domain $\Omega = [-2.5, 2.5]^2$, $u, v$ being the velocities in the horizontal and vertical direction, $h$ describing the water depth and $b$ describing a spatially varying bathymetry. $hu, hv$ can be interpreted as

Table 7: Architectural hyper-parameters for FNO and DKFNO. For a fair comparison, they have the exact same hyper-parameters.

|  | Width | Modes | Fourier Layers |
|---|---|---|---|
| Navier Stokes | 20 | 12 | 4 |
| Darcy Flow | 32 | 12 | 4 |
| Shallow Water | 20 | 16 | 4 |

Table 8: Training hyper-parameters for all models.

| Hyper-parameter | Value |
|---|---|
| Learning Rate | 0.001 for all except 0.0001 for GNO |
| Weight Decay | $1e-4$ |
| Scheduler | CosineAnnealingLR |
| Epochs | 500 for all except 2000 for DeepONet |
| Batch Size | 20 |

the directional momentum components and $g_r$ describes the gravitational acceleration. We are interested in learning the operator mapping the initial condition $h(t = 0, x, y)$ to a later time $h(t = T, x, y)$.

The specific simulation they include in their benchmark for the shallow-water equations problem is a 2D radial dam break scenario. On a square domain $\Omega = [-2.5, 2.5]^2$. The water height is initialized as a circular bump in the center of the domain

$$h(t = 0, x, y) = \begin{cases} 2.0, & \text{for } r < \sqrt{x^2 + y^2} \\ 1.0, & \text{for } r \geq \sqrt{x^2 + y^2} \end{cases}$$

with the radius $r$ randomly sampled from $\mathcal{U}(0.3, 0.7)$. For generating the datasets, they simulate this problem using the PyClaw (Ketcheson et al., 2012) Python package which offers a comprehensive finite volume solver.

### F.2 Model and Training Details

We adhere to the FNO convention by omitting a validation set and training for a fixed number of epochs without early stopping. However, for CNO and GT, we include a validation set and select the best model based on the validation set, as we observe that the errors fluctuate more for these two baselines. All experiments are repeated 10 times (except for the Navier-Stokes equation with $\nu = 1e-4$) to reduce randomness and ensure robustness of the results. For each baseline comparison, we provide the details as follows:

- FNO (Li et al., 2021): Adopted from the code provided by the authors in their GitHub repository with no change to their architecture.

- FFNO (Tran et al., 2023): Adopted from the code provided by the authors in their GitHub repository with no change to their architecture.

- GT (Cao, 2021): We re-implemented the Galerkin Transformer using the authors' implementation from their GitHub repository. Since the PDE problems considered in our work are on uniform grids, we chose the identity mapping instead of GNNs as encoders for efficiency.

- ONO (Raonic et al., 2023): Adopted from the code provided by the authors in their GitHub repository with no change to their architecture.

- CNO (Raonic et al., 2023): Adopted from the code provided by the authors in their GitHub repository with no change to their architecture.

- U-Net: Adopted from the code provided by PDEBench (Takamoto et al., 2023) with no change to their architecture at all.

- DeepONet (Lu et al., 2021): Adopted from the Torch version provided by the authors in their GitHub repository. Specifially, for the Branch network, we have tested both MLP and CNN-based network. We found that CNN-based network yields better performance, so we chose a CNN-based network as the Branch. An MLP is used as the trunk network.

- DSFNO: For a fair comparison, DSFNO follows exactly the FNO architecture, except that the Fourier convolution layers with fixed Fourier filters are replaced by the dynamic kernels produced by the efficient Schwartz Kernel Operator described in Sec. 4.3.

**Architectural Hyperparameters.** All the baseline models are adopted from their original work with no change at all to the architectural hyperparameters. We report the hyperparameters for FNO and DSFNO in Table 7. Note that the hyperparameters for the Navier-Stokes and Darcy flow equations are followed exactly from the original work (Li et al., 2021) as well.

**Training Hyperparameters.** We report the training hyperparameters in Table 8.

