# OpenReview forum: "Dynamic Schwartz-Fourier Neural Operator for Enhanced Expressive Power"
_TMLR — Accepted by TMLR_

### Review · Reviewer_ct7k · 2025-02-20

**Summary Of Contributions:**

The authors in this paper introduces the Dynamic Schwartz-Fourier Neural Operator (DSFNO), an enhancement to Fourier Neural Operators (FNOs) with two key innovations that I think are worth highlighting:
1. Dynamic kernels that adapt to input functions, rather than using fixed convolution kernels as in standard FNOs. This allows the model to better handle systems lacking translational symmetry.
2. Schwartz operators that enable interactions between different frequency modes, expanding beyond the point-wise multiplication in frequency domain used in FNOs.

The authors also provide (Some) theoretical analysis showing that:
1. Traditional FNO activation functions induce limited frequency domain interactions that preserve translation equivariance
2. Any integral operator can be expressed as frequency domain interactions through the Schwartz kernel theorem
3. An efficient implementation of frequency mode interactions can reduce complexity from O(k⁴) to O(k³)

Overall its a cool work: )!

The empirical evaluation demonstrates improved performance over FNO and other baselines on three PDE problems:
1. Incompressible Navier-Stokes equation
2. Darcy flow equation
3. Shallow water equation

**Audience:**

Yes

**Claims And Evidence:**

Yes

**Requested Changes:**

Maybe some changes (again not required all of them but it would be great if most of them are incorporated)
1 - Show some visualizations of how learned kernels vary with different input functions
2 - Include failure case analysis - when might fixed kernels actually be preferable?
3 - Add detailed complexity analysis including memory requirements and training time comparisons across methods/
4 - Analyze stability of dynamic kernel generation
5 - Better explain the connection between Schwartz operators and frequency interactions
6 - Clarify how mode truncation affects dynamic kernel learning, again theres lots of works out there understanding the spectral bias of FNO in particular. How would that change this? There is a recent work called Incremental FNO https://openreview.net/forum?id=xI6cPQObp0 which exploits this.
7 - Compare to the amortized version as mentioned in the weakness.
8 - Does not clearly explain the architecture of their specific kernel generator? Like what has been used? Showing ablation studies with different generator architectures would be nice.
9 - Ablation studies in the weakness.

**Strengths And Weaknesses:**

Strengths:
For me the biggest strengths is that the authors present a rigorous theoretical foundation, carefully connecting the proposed methods to fundamental concepts in functional analysis and operator theory. The authors provide clear mathematical justification for their approach through the Schwartz kernel theorem and frequency domain analysis. The experimental evaluation is comprehensive, testing across multiple challenging PDE problems with different physical characteristics (thank you for doing 10 runs). The ablation study is particularly well done, systematically demonstrating the contribution of both the dynamic kernel and frequency interaction components. The paper also presents an efficient implementation strategy that keeps computational complexity manageable, showing practical consideration beyond just theoretical improvements.

But I have several points of weaknesses that needs to be clarified

The paper's central claim about the benefits of adaptive kernels requires more thorough justification and analysis. While the authors argue that fixed convolution kernels in FNOs limit expressivity, especially for systems lacking translational symmetry, they don't provide convincing theoretical or empirical evidence for why learning input-dependent kernels necessarily solves this limitation. The mechanism by which the dynamic kernel actually captures relevant patterns in the input function remains somewhat unclear. The authors show improved performance but don't adequately explain the underlying reason - is it truly because of better handling of spatial dependencies, or could it be simply due to increased model capacity? For me its the latter reason as you just are learning the frequency interactions but there have been work doing that especially this work [1] https://openreview.net/attachment?id=a6em980M9x&name=pdf where they amortize the FNO in the Spectral block using MLP/KAN instead of just doing a simple linear transformation.

Again I do like the results, but while they do show improvement over baselines, they don't demonstrate the kind of dramatic performance gains one might expect from a fundamental architectural innovation. This raises questions about whether the added complexity of dynamic kernel generation is truly justified. The paper also lacks detailed analysis of the computational overhead introduced by the neural kernel generator, including memory requirements and training stability considerations.

Also theoretically all neural operator models are learning the kernel that is a good approximation to the greens function for the PDE. So for a fixed PDE it makes sense to learn the shared solution operator over different initial conditions for that single PDE. So not sure why one would try to learn a dynamic kernel a lot more, this makes sense however when we move on to training foundation models say pretraining over different PDEs? Again I maybe mistaken here but atleast my intuition says that the advantage of a dynamic kernel isnt that much to learning just a static one for a fixed PDE problem ( like the authors have done in this paper, all single PDE experiments).

Lastly, it would be great to include ablation studies of how by varying the number of modes, how the architecture performs. The reason the simple FNO works so well is that the linear transform basically zeros out all the high frequency components, but in the paper that is proposed here, all these are based to the generator and mixed, so you could overfit on the noise etc?

---

> ### Author Response · Authors · 2025-03-17
> **Reply to Reviewer ct7k Part I**
>
> Thank you so much for taking the time to review our work. We are glad that you feel "Overall its a cool work: )". Your insights and feedback are greatly appreciated. We have made necessary changes based on your comments. We have revised the paper as you requested. All the changes are marked in red for your convinience. In addition, we address your comments below.
>
> > W1: they don't provide convincing theoretical or empirical evidence for why learning input-dependent kernels necessarily solves this limitation.
>
> - **We have revised the paper to include a more rigorous discussion.** In Sec. 4.2, under "Discussion on Translation Equivariance", we refer to readers to Appendix C.2 for a detailed analysis on why frequency domain interactions can remove the unwanted translation symmetries. **We also provide here the high-level ideas for your convinience**:
>   - At a high level, the simplest frequency domain interaction can be seen as just a weighted summation between the frequencies; if it is a more complicated one, it will be multiple of such linear layers with non-linear activation functions such as a Spectral neural operator. **A weighted summation in the frequency domain is no longer a convolution but a full integral $\int_\Omega k\left(x,y\right) v(y) dy$ in the real space** (see Theorem 4.3, the proof works both ways). Thus, it is naturally not equivairant. From the frequency domain perspective, mixing Fourier modes will break the phase shift property, thus violating the Fourier shift theorem.
>
> > W2: The mechanism of dynamic kernel, improvement with increased model capacity, comparison with amortized FNO.
>
> - **The key advantages of dynamic kernel can be summarized twofolds**:
>   1. The ability to adapt to different input functions, which conforms to that of the Green's function kernels.
>   2. As discussed in Sec. 4.2, the model gains greater expressive power with the introduction of frequency domain interactions. Furthermore, in Sec. 4.3, we present the formulation of the Schwartz Kernel Operator, which models a full integral rather than just a convolution. It is evident that full integral operators are more expressive, as convolution operators represent a special case of integral operators.
>
> - **It is not due to increased model capacity.** The number of parameters barely increases from $0.93$ million to $1.06$ million. In the table below, we additionally present the results when we enlarge the FNO models to have a larger channel width so that the number of parameters are on par with those of DSFNO. Clearly, we observe that increasing the model capacity of FNO does not lead to a significant performance gain, as demonstrated by DSFNO.
>
> |  | #Par | Navier-Stokes 1e-3 |
> | :- | :- | :- |
> | FNO | 0.93 | $0.795 \pm 0.07$ |
> | FNO more channels | 1.13 | $0.793\pm 0.06$ |
> | DSFNO (ours) | 1.06 | $0.562 \pm 0.04$ |
>
> - **Our work and pespective are completely different from that of the Amortized FNO (AM-FNO [1]).**
>   - AM-FNO amortizes the parameters of the Fourier filters by using an MLP (either KAN or a standard MLP). To summarize AM-FNO, the total number of parameters is $C_{\text{in}} \times C_{\text{out}} \times N^d_{\text{modes}}$, where $d$ is the dimension of the domain. AM-FNO amortizes these parameters using an MLP that takes the frequencies as input and produces the coefficient values for all channels. Therefore, the complexity is reduced to $O(C_{\text{in}} \times C_{\text{out}})$ as a result of the amortization on the Fourier modes.
>   - This is completely different from our method (DSFNO). First of all, our kernel is adaptive to the input function, where as AM-FNO is not. Secondly, AM-FNO do not invoke frequency domain interaction (FDI) at all as the armotization network takes fixed frequencies as input, not the input parametric function of the operator learning task. In our work, the Neural Kernel Generator takes the input parametricfunction to produce a kernel, which involves FDI of the input function. Note that in the first paragraph under Definition 4.1, we state "FDI means that the update of one Fourier coefficient is dependent on the other Fourier coefficients". **Clearly, AM-FNO just armotizes the network parameters without any consideration of the input, the update of one Fourier coefficient is still only dependent on one Fourier coefficient. There is no FDIs in AM-FNO.**
>
> >W3: Computational overhead.
>
> - In Sec. 4.3, we have discussed the complexity of DSFNO, $O(k^2)$ with $k \leq n$ being the number of truncated modes, against that of FNO, $O(n \log n)$. Note that with a small $k$, $k^2$ might be smaller than $n \log n$, and in that case, DSFNO is still subject to the complexity of FFT, which is $O(n \log n)$.
>
> - **In appendix F.1.1, we have additionally included the experimental results on the training time and memory requirements for FNO and DSFNO, respectively.** DSFNO only introduces a little overheads(18.76 seconds per epoch for FNO, 21.59 for DSFNO; memory requirement almost the same).

---

> > ### Author Response · Authors · 2025-03-17
> > **Reply to Reviewer ct7k Part II**
> >
> > > W4: Approximation to greens function for the PDE, foundation model across different PDEs
> >
> > - There could be some misunderstanding here. The learning task is for a family of parametrized PDEs. In certain cases, there could be a fixed Green's kernel for the underlying learning problem; **however, in many cases, there is no fixed Green's kernel**. For example, in the Darcy flow example, the Green's kernel actually depends on the coefficient function $a(x)$, and the task is to map coefficient functions to solution functions. Moreover, in many complex PDEs, a Green's kernel may not even exist. We use the Green's function perspective as a motivation in our work, just as the Green's kernel motivates the kernel integral operator framework [2, 3].
> >
> > - While we do believe that dynamic kernels might offer great potential in learning a foundation model, it is beyond the scope of the current work. We have included this perspective in our discussion of future works.
> >
> > >W5: Truncation of Fourier modes
> >
> > - There could be some misunderstanding here. **In the DSFNO framework, we zero out the high frequency components as well. In fact, we follow the exact same truncation as in the FNO architecture.** The generator only mixes the frequencies within the truncation. This is stated in Sec. 4.3 of the paper:
> >   - *Similar to the truncation used in FNO, we can restrict the interaction to a subset of low-frequency modes, yielding a complexity of $O(k^2)$, where $k$ is the number of truncated Fourier modes.*
> >
> > - Given the clarification above, we are not sure if the ablation study on varying the number of modes is still necessary to include. We present the results below. However, if you find it necessary to include these results in the paper, we will definitely not hesitate to include.
> >
> > **Requested Changes**
> >
> > > C1: More visualization of learned kernels
> >
> > - **We have included more visualizations in Appendix A.**
> >
> > > C2: Include failure case analysis
> >
> > - As we have emphasized in Sec. 5, "There is no single state-of-the-art neural operator in the domain of neural PDE solvers." Our models can be outperformed on certain datasets by other methods, which is common in operator learning. **Failure analysis of certain neural operators is difficult to conduct and should be considered in the research direction focused on explaining how neural operators learn (XAI).**
> >
> > > C3: Runtime and memory comparison
> >
> > - In Sec. 4.3, we have discussed the complexity of DSFNO, $O(k^2)$ with $k \leq n$ being the number of truncated modes, against that of FNO, $O(n \log n)$. Note that with a small $k$, $k^2$ might be smaller than $n \log n$, and in that case, DSFNO is still subject to the complexity of FFT, which is $O(n \log n)$.
> >
> > - **In appendix F.1.1, we have additionally included the experimental results on the training time and memory requirements for FNO and DSFNO, respectively. DSFNO only introces a little overhead**.
> >
> > >C4: Stability of dynamic kernel generation
> >
> > - **For the specific efficient Schwartz Kernel Operator we use, it is stable for the numerical tests we conduct.**
> >
> >
> > >C5: Schwartz operators and frequency interactions
> >
> > - As shown in Theorem 4.3, the Schwartz operator can be realized as a kernel integral (matrix-vector multiplication in the discrete case). Naturally, a kernel integral would involve all the Fourier modes. In contrast, a convolution operator is a point-wise multiplication in the frequency domain.
> >
> > - **We have included an explanation in the paragraph under Theorem 4.3 to better clarify the connection between the Schwartz operator and frequency interactions.**
> >
> > >C6: How mode truncation affects dynamic kernel learning
> >
> > - Clearly, any truncation would result in some loss of information. However, the common assumption in FNO is that we use a large enough truncation threshold such that the information loss is nearly negligible. Even the universal approximation property of FNO is contingent upon this [4].
> >
> > - Similarly, for DSFNO, a reasonable truncation threshold must be chosen to retain the information of the original signal. Otherwise, truncation will naturally impact learning for both FNO and DSFNO. Please refer to our responses to the 5th weakness, **DSFNO follows exactly the same truncation as FNO for a fair comparison.** This is already discussed in Sec. 4. **We have also revised the paper to include this information in the architectural details in Appendix F.2.**

---

> > > ### Author Response · Authors · 2025-03-17
> > > **Reply to Reviewer ct7k Part III**
> > >
> > > > C7: Comparison with AM-FNO
> > >
> > > - **As discussed in the response to this weakness, the scope of AM-FNO is completely different from that of ours.** We include a discussion of this work in the related work as an important improvement to FNO. However, if you feel it is still necessary to compare our method with AM-FNO, please let us know, and we will update it in the next revision.
> > >
> > > > C8: Specific kernel generator used
> > >
> > > - The efficient Schwartz Kernel Operator is used. **We have included more details in Appendix F.2 (Model and Training Details).** We would like to perform an ablation study on different kernel generators; however, the computational costs would be prohibitive.
> > >
> > >
> > > > C9: Ablation on the truncation of modes
> > >
> > > - As discussed in our response to the weakness, DSFNO also truncates the Fourier modes, and this is stated in Sec. 4.3. All the experiments are done with the same truncation mode as FNO. **We have revised the paper to include the details in Table 6 in Appendix F.2.** With this clarification, if you still want to see such an ablation study, please let us know, and we will include it in the next revision.
> > >
> > > [1] Amortized Fourier Neural Operators, Zipeng Xiao et al, NeurIPS 2024
> > >
> > > [2] Neural Operator: Graph Kernel Network for Partial Differential Equations, Zongyi Li et al
> > >
> > > [3] Fourier Neural Operator for Parametric Partial Differential Equations, Zongyi Li et al., ICLR 2021
> > >
> > > [4] On Universal Approximation and Error Bounds for Fourier Neural Operators, JMLR, 2021

---

> > > > ### Comment · Reviewer_ct7k · 2025-03-20
> > > > **Response**
> > > >
> > > > Based on the revised paper and the author's response to my concerns, I believe they have addressed most of the major issues that I raised.I am happy with the revised work and the answer to all my questions: )

---

### Review · Reviewer_1C32 · 2025-02-23

**Summary Of Contributions:**

This paper presents DSFNOs to enhance the expressive capability of FNOs. Specifically, the authors argue the translation-invariant design of FNO and propose an efficient way to include interactions into the projection process of the frequency domain, which is accomplished by the adaptive kernel. Experimentally, DSFNO surpasses FNO in solving Shallow Water, Darcy Flow and Navier-Stokes equations with only a little extra computation.

**Audience:**

Yes

**Broader Impact Concerns:**

This paper discusses the deficiency of FNO, which can be inspiring for future research.

**Claims And Evidence:**

Yes

**Requested Changes:**

1. Discuss the frequency domain interaction under the context that elements in the frequency domain are orthogonal.

2. Compare with more recent Transformer-based baselines.

3. Give more details for Fig 1.

**Strengths And Weaknesses:**

## Strengths
1. The analysis of the weakness of the current design in FNO is reasonable and inspiring.

2. This paper is well-written and the overall performance looks convincing.

## Weaknesses

1. The motivation for including frequency domain interaction is not clear enough.

I am convinced by the argument that FNO needs adaptive kernels, but why does DSFNO conduct interaction in the frequency domain? It should be noticed that different elements in the frequency domain actually represent orthogonal bases, which means that they contain less relative information. I cannot understand the advantages of this operation. Besides, if you delve into the amplitude in the frequency domain, you may find that the low-frequency values will be much larger than the high-frequency ones. It may be meaningless to include the high-frequency components in the interaction. Thus, I think more ablation studies or discussions should be presented.

2. More baselines are expected.

Although FNO is a classical method in this community, many Transformer-based models achieve better performance, such as Transolver [1] and Factformer [2]. It should be considered to add them to the experiments or at least discuss the relation w.r.t. these new Transformer-based models.

[1] Transolver: A Fast Transformer Solver for PDEs on General Geometries, ICML 2024
[2] Scalable Transformer for PDE Surrogate Modeling, NeurIPS 2023

3. About the visualization in Fig 1.

In general, I appreciate the visualization of learned filters. However, what are these visualizations actually about? I think the authors should provide more implementation details of the visualization process to make readers understand the meaning of each filter.

---

> ### Author Response · Authors · 2025-03-17
> **Reply to Reviewer 1C32 Part I**
>
> Dear Reviewer 1C32, thank you for taking the time to review our work and provide valuable feedback. We have carefully revised the paper, with changes marked in red, and address your comments below.  We also noticed that **the "Claims and Evidence" criterion is currently marked as "No." If our responses clarify your concerns, we kindly ask you to consider updating your review.**  If you have further concerns, please do not hesitate to let us know, and we will do our best to address them.
>
> > W1: The motivation for including frequency domain interaction is not clear enough.
>
> **Overall, we think there might be some misunderstanding here.** We discuss the motivation behind frequency domain interactions (FDIs) in Sec. 4.2; specifically, the sentence in bold under Theorem 4.2 emphasizes that FDIs invoked by activation functions are restricted. Moreover, in Sec. 4.3, we provide Theorem 4.3, which states that a linear layer in the frequency domain corresponds to a full integral in real space, further justifying the need for frequency domain interactions. **Additionally, we have modified the paper to include a sentence in Sec. 4.3 to explicitly emphasize the direct benefits of FDIs through Schwartz kernels.**
>
>
> We respond to your comments in more details below:
>
> - First, **while Fourier modes are independent during the compression process, this does not imply that they should be treated totally independently in the learning process.** For instance, recovering the full information of the original signal requires all frequency components (or at least the low frequencies for a reasonable approximation).
>   - As a specific counterexample, in FNO, the spectral convolution is point-wise multiplciation in the frequency domain, which does not involve frequency domain interaction (FDI). However, this can obviously **only learn linear operators**. To break non-linearity, non-linear activation functions are included, which actually involves **limited** frequency domain interactions. At a high-level, non-linear activation functions can only involve frequency domain interactions, subject to maintaining all the Euclidean equivariance in the physical space. However, such equivariance is not always wanted, and **our intentionally designed FDI is not limited and can be more expressive than those represented by activation functions.**
>
> - **We argue that frequency domain interactions are essential for learning more diverse spatial patterns as they can represent the full kernel integral.** As stated in Theorem 4.3, a simple frequency domain interaction, when implemented as a linear layer on the frequencies, is mathematically equivalent to a **full kernel integral** rather than a convolution integral in FNO. Since **convolution integrals are a special case of kernel integrals**, this suggests that frequency domain interactions can enhance the expressiveness of FNO. Moreover, even **from the perspective of Green's kernels, the kernels do not have to be convolutional kernels; oftentimes, they are just general integral kernels.**
>
> - Last but not least, we totally agree with you that high-frequency components are meaningless if the signals (input and output functions) do not have high-frequency information. **We indeed zero-out the high-frequency components.** This is already described in the paper and also implemented in all of our experiments. We provide quotes from the paper here for your convenience:
>   - In the second paragraph in Sec. 4.1, we state that "In practice, the Neural Kernel Generator is a transformation mapping **truncated Fourier coefficients** of the input function to a band-limited output convolution kernel function."
>   - Moreover, we have emphasized several times that we can use the **truncated Fourier coefficients** in the paper. For example, in the second paragraph in Sec. 4.2 (the second sentence above Theorem 4.2) and in the second sentence in Sec 4.3.
>   - Moreover, in the complexity analysis of the Schwartz Kerenl Operator, which is $O(k^2)$, we state that "$k$ is **the number of truncated Fourier modes**."

---

> ### Author Response · Authors · 2025-03-17
> **Reply to Reviewer 1C32 Part II**
>
> >W2: Transformer-based baselines
>
> - **We found that transformer-based architectures may not perform well on datasets where FNO excels, such as the Navier-Stokes dataset with a low Reynolds number.** For example, transformer-based models usually perform well on the Navier-Stokes example with a viscosity constant of 1e-5; however, we provide the performance of Transolver [2] on the dataset with a viscosity constant of 1e-3 below. Transolver's performance is very poor on this particular example. **We have experimented with various transformer-based architectures on this problem ($1e{-3}$); none of them yields satisfying results.** This could be attributed to the difference in learning mechanisms between frequency domain learning (FNO and our proposed DSFNO) and physical domain learning (transformer-based approaches). **The learning pattern in this dataset may appear to be more prominent in the frequency domain** (in fact, this PDE is solved using a pseudo-spectral method). Additionally, the error compounds across auto-regressive time steps, making it appear more significant. **The Navier-Stokes equations with a viscosity constant of $1e{-3}$ and $1e{-4}$, using the exact same setting as in FNO [1], are not reported in most transformer-based architectures, including the provided references.** However, transformer-based architectures are still very important in many other tasks; we discuss this in Appendix E of the revised paper.
>
> |  | Navier-Stokes 1e-3 |
> | :--- | :--- |
> | FNO | $0.795 \pm 0.07$ |
> | Transolver | $77.792 \pm 19.19$ |
> | DSFNO (ours) | $0.562 \pm 0.04$ |
>
>
>
> - **The most valuable comparison for DSFNO is against FNO.** FNO and DSFNO tend to perform better than transformer-based models when global patterns are more prominent due to their ability to efficiently and accurately capture such patterns. However, they can be prone to over-smoothing when small-scale features are more significant, whereas transformer-based models excel in such tasks. Such issues with FNO have been noted and can be mitigated in [1]. The same approch can be taken for our proposed DSFNO as well; however, it is beyond the scope of our work.
>
> - **We have included detailed discussions about this learning difference and noted these two works (Transolver and Factformer) as pioneering works in transformer-based neural operators in Appendix E, with initial discussions under Baselines in Sec. 5 in the main paper.**
>
> > W3: More details about Fig. 1.
>
> - **We have included a more detailed description for Fig. 1 in the paper and provide our response here.** This figure shows the learned filters in one kernel integral layer of FNO and DSFNO for the Darcy flow equation. Each layer contains in_channels $\times $ out_channels filters. We visualize the filters for the in_channels in the first kernel integral layer in both FNO and DSFNO, respectively.  We want to emphasize two key observations from this visualization:
>    - The DSFNO filters are adaptive and change with the input function.
>    - These adaptive filters allow the model to focus on different parts of the input to extract meaningful features.
>
> > Requested Changes
>
> All the requested changes are essentially the same as the weaknesses. Please refer to the answers above.
>
> [1] Neural Operators with Localized Integral and Differential Kernels, Miguel Liu-Schiaffini el al., ICML 2024
>
> [2] Transolver: A Fast Transformer Solver for PDEs on General Geometries, Haixu Wu et al., ICML 2024

---

> > ### Comment · Reviewer_1C32 · 2025-03-18
> >
> > Thanks for your response. My concerns about the motivation of FDI have been resolved. Thus, I have changed the "Claims And Evidence" to Yes.
> >
> > About the performance of Transolver, I think it is kind of strange as Transolver performs quite well in Navier-Stokes 1e-5. For double check, most Transformer-based models are under the strategy of teacher forcing. I would suggest the authors rerun this experiment to avoid potential misleading results.

---

> > > ### Author Response · Authors · 2025-03-19
> > > **Second Round Reply to Reviewer 1C32**
> > >
> > > Thank you so much for your rapid response. We are glad to know that our clarification resolves your concerns on the need of FDIs.
> > >
> > > >Regarding the performance of transolver
> > >
> > > - **In fact, we used the original setup directly (teacher forcing is adopted in their original setup).** The training error is very small (~$0.3$%); it suffers significantly from overfitting issues. We have tried different approaches such as adding regularization techniques (e.g., weight decay, dropout), tuning the learning rate schedule, and early stopping, but the model still fails to generalize well on this dataset. **We have also tried training without the teacher forcing strategy; however, the training is very unstable and performance is still not satisfying (large variance, with the best $\ell_2$ error across 3 runs being around $56$%).**
> > >
> > > - **Regarding the outperformance on 1e-5, the change in viscosity actually affects the dataset and the learning pattern.** First, as the viscosity decreases (Reynolds number increases), the data exhibits more small-scale turbulence and high-frequency variations that are more apparent in the physical space; therefore, it is more suitable for physical space models such as transformer-based. This is why transformer-based models can perform well on the 1e-5 dataset. However, **the learning pattern in the 1e-3 dataset may appear to be more prominent in the frequency domain (in fact, this PDE is solved using a pseudo-spectral method).**
> > > Hence, for this dataset, FNO-based models excel but transformer-based models usually yield poor results.
> > > Second, there are only $10$ auto-regressive prediction steps for 1e-5; there are $40$ prediction steps for 1e-3, and transformer-based architectures might struggle with long-time rollouts.
> > >
> > > - **Regarding the outperformance of Transolver**, Transolver adopts the teacher forcing strategy, whereas in their paper, the results for FNO are taken from the original FNO paper, which does not use teacher forcing. As noted in [1], the teacher forcing strategy will improve the performance on auto-regressive tasks. Lastly, the FNO architecture has been updated (with generally better performance) since the publication of their paper. In our paper, we report the updated results (for example, $0.795$% on 1e-3 while the original reported error in [2] is $1.28$%).
> > >
> > >
> > > - We understand your concerns that this could be misleading, and that is why we do not include them in the revision. We do believe that transformer-based architectures, such as Transolver and Factformer, are very important and effective in many complex problems. Unfortunately, it does not perform well in this case, but it should not undermine the importance of transformer-based models. **The development of transformer-based models and FNO-based models are both highly important.** As noted in our first round of response as well as in the paper (Appendix E), there is a difference in learning mechanisms between frequency domain learning (FNO and our proposed DSFNO) and physical domain learning (transformer-based approaches). Transformer-based models are more suitable and effective in learning patterns that are more prominent in the physical space.
> > >
> > >
> > > [1] Factorized Fourier Neural Operators, Alasdair Tran et al., ICLR 2023
> > >
> > > [2] Fourier Neural Operator for Parametric Partial Differential Equations, Zongyi Li et al., ICLR 2021

---

### Review · Reviewer_ebVa · 2025-03-10

**Summary Of Contributions:**

This Paper proposes a dynamic Schwartz operator that induces interactions between modes to enhance the expressiveness of FNOs. It equips FNOs with Schwartz operators to learn dynamic kernels, termed Dynamic Kernel Fourier Neural Operators (DSFNOs). By incorporating this dynamic mechanism, our model gains the ability to capture relevant frequency information patterns, facilitating a better understanding and representation of complex physical phenomena.

**Audience:**

Yes

**Claims And Evidence:**

Yes

**Requested Changes:**

1) The idea is quite similar to the following paper [1]. Can you describe the advantages it offers compared to [1]?

[1] Xiao, Z., Kou, S., Zhongkai, H., Lin, B., & Deng, Z. (2024). Amortized Fourier Neural Operators. Advances in Neural Information Processing Systems, 37, 115001-115020.

2) How does the computational complexity (training time) change with the dynamic kernel?

3) In FNO, modeling the Navier-Stokes equations with a viscosity coefficient of 1e-5 is more challenging compared to 1e-4 and 1e-3. Why haven’t you included it as a benchmark as well?

4) The complete architectural details are not provided in either the main paper or the supplementary material.

5) How does the model perform on the super-resolution task? Have you conducted experiments on it?

**Strengths And Weaknesses:**

**Strength:**
1) Dynamic Kernel Integral enhancing the expressive power of fixed convolution kernels in FNO.

2) Schwartz operators that represent general linear operators between function spaces while facilitating dimensionality reduction for efficiency.

**Weakness:**

1) "There is no single state-of-the-art neural operator in the domain of neural PDE solvers." Proposed method is not compared with latest baselines such as LSM, Transolver, Amortized FNO etc.

[1] Wu, H., Hu, T., Luo, H., Wang, J., & Long, M. (2023). Solving high-dimensional pdes with latent spectral models. arXiv preprint arXiv:2301.12664.

[2] Wu, H., Luo, H., Wang, H., Wang, J., & Long, M. (2024). Transolver: A fast transformer solver for pdes on general geometries. arXiv preprint arXiv:2402.02366.

[3] Xiao, Z., Kou, S., Zhongkai, H., Lin, B., & Deng, Z. (2024). Amortized Fourier Neural Operators. Advances in Neural Information Processing Systems, 37, 115001-115020.

2) Hyperparameter details, architectural details and experimental details are not provided.

---

> ### Author Response · Authors · 2025-03-17
> **Reply to Reviewer ebVa Part I**
>
> Dear Reviewer ebVa, thank you for your constructive comments. We have revised the paper (marked in red) based on your suggestions, and we provide the responses below.
>
> >W1: Comparison with more baselines
>
> - **We have a recent baseline, ONO [1], which was published at ICML 2024, concurrent with the provided references.** However, we completely understand your concerns. **We have additionally conducted further experiments with the latest baselines Transolver [2] and AM-FNO [3] on the Navier-Stokes equation with a viscosity constant of $1e{-3}$**. The results are presented below. Transolver suffers from significant overfitting issues, which is magnified through the autoregressive time steps. We have tried different approaches to mitigate; however, none of them works. The Navier-Stokes equations with a viscosity constant of $1e{-3}$ and $1e{-4}$, using the exact same setting as in FNO [4], are not reported in most transformer-based architectures, including the provided references. Given the instability of transformer-based architectures on the Navier-Stokes equation with low Reynolds number, we do not include them in the revision. If you feel it is necessary, please let us know; we will include them in the next revision. For AM-FNO, please refer to our response for the first requested change.
>
> |  | Navier-Stokes 1e-3 |
> | :--- | :--- |
> | FNO | $0.795 \pm 0.07$ |
> | AM-FNO | $1.296 \pm 0.03$ |
> | Transolver | $77.792 \pm 19.19$ |
> | DSFNO (ours) | $0.562 \pm 0.04$ |
>
> Transolver is averaged over 5 runs due to a larger computational cost of the transformer architecture while all others are over 10 runs.
>
> - **We acknowledge that all three works are highly insightful and groundbreaking, and we have discussed them in Appendix E with initial discussion in Sec. 5 under "Baselines".** Transformer-based neural operators and FNO-based neural operators represent two key branches in the field. Each can outperform the other on a given dataset, but may be outperformed in turn on different datasets. Therefore, both approaches are of equal importance. Our work focuses on advancing FNO-based neural operators.
>
> > W2: Hyperparameter details, architectural details and experimental details are not provided.
>
> - **The experimental details are provided in Appendix F. The architectural details of the baselines are provided in Appendix F.2.** The DSFNO architecture is implemented as described in Sec. 4.3 (Efficient Schwartz Kernel Operator). The architectural hyperparameters for the baselines follow their original works, and those of the DSFNO are set to be the same as those of FNO for a fair comparison.
>
> - **We have included additional information and tables in Appendix F.2 to help readers better understand the report of these details.**
>
> Requested Changes:
>
> > C1: Comparison with AM-FNO
>
> - **Our work and pespective are completely different from that of the Amortized FNO (AM-FNO [3]).**
>   - AM-FNO amortizes the parameters of the Fourier filters by using an MLP (either KAN or a standard MLP). To summarize AM-FNO, the total number of parameters is $C_{\text{in}} \times C_{\text{out}} \times N^d_{\text{modes}}$, where $d$ is the dimension of the domain. AM-FNO amortizes these parameters using an MLP that takes the frequencies as input and produces the coefficient values for all channels. Therefore, the complexity is reduced to $O(C_{\text{in}} \times C_{\text{out}})$ as a result of the amortization on the Fourier modes.
>
>   - This is completely different from our method (DSFNO). First of all, our kernel is adaptive to the input function, where as AM-FNO is not. Secondly, AM-FNO do not invoke frequency domain interaction (FDI) at all as the armotization network takes fixed frequencies as input, not the input parametric function of the operator learning task. In our work, the Neural Kernel Generator takes the input parametricfunction to produce a kernel, which involves FDI of the input function. Note that in the first paragraph under Definition 4.1, we state "FDI means that the update of one Fourier coefficient is dependent on the other Fourier coefficients". **Clearly, AM-FNO just amortizes the network parameters without any consideration of the input. The update of one Fourier coefficient is still only dependent on one Fourier coefficient. There is no FDIs in AM-FNO.**
>
> - **As the scope of our work and that of the AM-FNO is totally different, we do not include a discussion of comparison in the revision; however, as an important improvement to FNO, we have included AM-FNO in related work.** Moreover, the same approach can be taken in DSFNO to amortize DSFNO as well to lessen the exponential growth of network parameters. If you think it is necessary to discuss this in more detail in the paper, please let us know, and we will include it in the next revision.

---

> > ### Author Response · Authors · 2025-03-17
> > **Reply to Reviewer ebVa Part II**
> >
> > > C2: Training time of DSFNO
> >
> > - We have already shown that DSFNO barely increases the inference time in Tables 1,2,and 3 in the paper. For the training time, we provide a comparison below between DSFNO and plain FNO as an example: On the Navier Stokes example with a viscosity constant of 1e-3 (Resolution $64 \times 64$ and $40$ roll-out times steps in total), **an epoch would take 18.76 seconds for FNO and 21.59 seconds for DKFNO, respectively. The peak memory usage of FNO is 12,034 MiB and that of DKFNO is 12,238 MiB. It is clear that DSFNO, using the efficient Schwartz Kernel Operator in Sec. 4.3, only adds a fraction of the computational hurdles in the training process.** We have included these results in Appendix F.1.1.
> >
> > > C3: Modeling the Navier-Stokes equations with a viscosity coefficient of 1e-5
> >
> > - **Operators that perform better on this dataset (viscosity constant 1e-5) do not necessarily perform better on "simpler" problems, such as that with a viscosity constant 1e-3.** AM-FNO outperforms FNO on the dataset with a viscosity constant 1e-5 (as shown in their paper) but not on the dataset with a viscosity constant 1e-3 (as shown above in our response to the first weakness). Therefore, we do not consider one problem to be more difficult than the other.
> >
> > - Transformer-based neural operators and FNO-based neural operators represent two key branches in the field. Each can outperform the other on a given dataset, but may be outperformed in turn on different datasets. Therefore, both approaches are of equal importance. **Our work focuses on advancing FNO-based neural operators.**
> >
> > - We acknowledge that FNO does not perform well on this dataset, potentially due to the turbulent characteristics and small-scale features under such a high Reynolds number. FNO might not capture these features effectively, as its global operations are often prone to oversmoothing and may fail to preserve local details [5]. DSFNO, being a global operation method, suffers from the same issue on this dataset. **There are ways to mitigate such issues [5], and they can be incorporated into our framework. However, this is beyond the scope of this work.**
> >
> > - **Taking all above into consideration, we do not include this specific example in our work.**
> >
> > > C4: The complete architectural details
> >
> > - DSFNO replaces the Fourier kernels in FNO with dynamic kernels produced by the efficient Schwartz Kernel Operator in Sec. 4.3. **We have included clearer descriptions in Appendix F.2 to better present the overall architectural and experimental details.**
> >
> > > C5: Superresolution tasks
> >
> > - In many recent works, the poor super-resolution performance of both FNO and transformer-based architectures has been noted [1,5,6]. It remains unclear whether zero-shot super-resolution is a meaningful performance evaluation metric, as even simple interpolation can outperform super-resolution [7]. Some recent studies [7,8] highlight that neural operators can be significantly affected by discretization mismatch errors. **Therefore, we do not conduct super-resolution tests or include super-resolution performance as an evaluation metric.**
> >
> > [1] Improved Operator Learning by Orthogonal Attention, Zipeng Xiao et al., ICML 2024
> >
> > [2] Transolver: A Fast Transformer Solver for PDEs on General Geometries, Haixu Wu et al., ICML 2024
> >
> > [3] Amortized Fourier Neural Operators, Zipeng Xiao et al, NeurIPS 2024
> >
> > [4] Fourier Neural Operator for Parametric Partial Differential Equations, Zongyi Li et al., ICLR 2021
> >
> > [5] Neural Operators with Localized Integral and Differential Kernels, Miguel Liu-Schiaffini el al., ICML 2024
> >
> > [6] Group Equivariant Fourier Neural Operators for Partial Differential Equations, Jacob Helwig et al., ICML 2023
> >
> > [7] Discretization-invariance? On the Discretization Mismatch Errors in Neural Operators, Wenhan Gao et al., ICLR 2025
> >
> > [8] Discretization Error of Fourier Neural Operators, Samuel Lanthaler et al.

---

> ### Comment · Action_Editor_ftCk · 2025-04-09
>
> Hi Reviewer ebVa,
>
>   It's been a while and all the other reviewers have submitted the final recommendation. Can you look at the authors' rebuttal and submit your final recommendation as well please? Thanks.
>
> AE

---

### Decision · Action_Editor_ftCk · 2025-04-26

**Recommendation:** Accept as is

**Comment:**

After the reviews and revision, all reviewers are happy about the paper being a meaningful contribution on top of FNOs.

**Audience:**

This is interesting to people interested in neural PDE solvers.

**Claims And Evidence:**

This Paper proposes a dynamic Schwartz operator that induces interactions between modes to enhance the expressiveness of FNOs. It equips FNOs with Schwartz operators to learn dynamic kernels, termed Dynamic Kernel Fourier Neural Operators (DSFNOs). By incorporating this dynamic mechanism, our model gains the ability to capture relevant frequency information patterns, facilitating a better understanding and representation of complex physical phenomena.